# Ezh2 phosphorylation state determines its capacity to maintain CD8[+] T memory precursors for antitumor immunity

Shan He[1], Yongnian Liu[1], Lijun Meng[1], Hongxing Sun[1], Ying Wang[1], Yun Ji [2], Janaki Purushe[3], Pan Chen[4], Changhong Li[4], Jozef Madzo[1], Jean-Pierre Issa[1], Jonathan Soboloff [1], Ran Reshef [5], Bethany Moore [6], Luca Gattinoni [2] & Yi Zhang [1,3]

Memory T cells sustain effector T-cell production while self-renewing in reaction to persistent antigen; yet, excessive expansion reduces memory potential and impairs antitumor immunity. Epigenetic mechanisms are thought to be important for balancing effector and memory differentiation; however, the epigenetic regulator(s) underpinning this process remains unknown. Herein, we show that the histone methyltransferase Ezh2 controls CD8[+] T memory precursor formation and antitumor activity. Ezh2 activates *Id3* while silencing *Id2*, *Prdm1* and *Eomes*, promoting the expansion of memory precursor cells and their differentiation into functional memory cells. Akt activation phosphorylates Ezh2 and decreases its control of these transcriptional programs, causing enhanced effector differentiation at the expense of T memory precursors. Engineering T cells with an Akt-insensitive Ezh2 mutant markedly improves their memory potential and capability of controlling tumor growth compared to transiently inhibiting Akt. These findings establish Akt-mediated phosphorylation of Ezh2 as a critical target to potentiate antitumor immunotherapeutic strategies.

[1] Fels Institute for Cancer Research and Molecular Biology, Temple University, Philadelphia, PA 19140, USA. [2] Center for Cancer Research, National Cancer Institute, Bethesda, MD 20892, USA. [3] Department of Microbiology & Immunology, Temple University, Philadelphia, PA 19140, USA. [4] The Division of Endocrinology and the Department of Pathology and Laboratory Medicine, The Children's Hospital of Philadelphia, University of Pennsylvania, Philadelphia, PA 19104, USA. [5] Columbia Center for Translational Immunology, Department of Medicine, Columbia University Medical Center, New York, NY 10032, USA. [6] Department of Internal Medicine, University of Michigan, Ann Arbor, MI 48105, USA. Correspondence and requests for materials should be addressed to S.H. (email: shan.he@temple.edu) or to Y.Z. (email: yi.zhang@temple.edu)

During an immune response, naive CD8$^+$ T cells (T$_N$) differentiate into short-lived effector cells, which eliminate the immediate threat, and long-term memory T cells, which are able to rapidly expand and elaborate effector function upon reencountering antigen[1–3]. T cells with impaired memory potential are unable to clear persistently expressed antigens, including those associated with chronic infections or tumors[2–9]. One important clinical scenario requiring optimal memory potential is adoptive cell therapy (ACT), where complete destruction of malignant cells requires maintenance of effector T cells (T$_{EFF}$) over weeks or months[7,8,10–12]. Akt activation drives CD8$^+$ T cells towards terminal differentiation and diminishes their memory potential[13]. Major transcription factors (TFs), including Id3, Id2, T-bet, Eomes and Blimp-1 control the differentiation and function of effector and memory cells[1,3]. Thus far, the mechanism by which CD8$^+$ T cells epigenetically integrate Akt signaling and these major TFs to regulate the memory potential for controlling tumor growth remains poorly defined.

Histone methylation regulates gene transcription patterns involved in multiple cellular processes[14–16]. In T cells, trimethylation of histone H3 at lysine 4 (H3K4me3) is enriched within gene promoters associated with active transcription. In contrast, H3K27me3 is deposited within gene loci correlated with repression of genes important for T-cell proliferation, differentiation and survival. Upon T-cell activation, the majority of these gene loci lose repressive H3K27me3 modifications, which is accompanied with initiating gene transcription[17–19]. Ezh2 together with Suz12 and Eed form core components of polycomb repressive complex-2 (PRC2) that catalyzes H3K27me3 and acts primarily as a gene silencer[16,20–22]. Intriguingly, Ezh2 has non-canonical activity in the regulation of signaling proteins and gene activation[23–27]. However, whether Ezh2 may regulate T-cell-fate decisions, and if this Ezh2 activity can be modified during T-cell responses, have not previously been determined.

Here we demonstrate that Ezh2 is essential for the development and maintenance of T memory precursors and associated antitumor immunity. Ezh2 activates Id3 while silencing Id2, Prdm1, and Eomes, promoting the expansion of memory precursors and their differentiation into functional memory cells. Akt mediates phosphorylation of Ezh2, which in turn eases Ezh2 transcriptional control, causing enhanced effector differentiation at the expense of memory precursors. Thus, Akt-mediated phosphorylation of Ezh2 may serve as a critical target to potentiate antitumor immunotherapeutic strategies.

## Results

**Ezh2 regulates effector and memory T-cell persistence.** To evaluate the impact of Ezh2 in CD8$^+$ T-cell responses, we used an experimental ACT model, of which transferred T cells were more effective in controlling tumor growth in lymphopenic mice compared with lymphoreplete hosts[4,11,28]. We deleted Ezh2 in melanoma-associated antigen gp100-specific CD8$^+$ T-cell receptor (TCR)-transgenic Pmel-1 cells by crossing Ezh2$^{fl/fl}$ CD4-Cre B6 mice[29] to Pmel-1 mice[30], which produced T-cell-specific Ezh2-knockout Pmel-1 mice (Ezh2$^{-/-}$ Pmel-1). Transfer of wild-type (WT) but not Ezh2$^{-/-}$ Pmel-1 T$_N$ repressed the growth of pre-established B16 melanoma in lymphodepleted mice (Fig. 1a). Using lymphopenic recipients without B16 tumor, we observed that the frequency and number of WT and Ezh2$^{-/-}$ Pmel-1 cells was similar in the spleen 4 days after transfer and immunization, while Ezh2$^{-/-}$ Pmel-1 cell numbers were decreased by 7 days and 35 days (Fig. 1b). The discrepancy was not due to differences in organ tropism, as fewer Ezh2$^{-/-}$ Pmel-1 cells were detected in peripheral blood (PB) and lymph node (LN) (Supplementary Fig. 1). Further, although Ezh2 deficiency did not affect the

capacity of Pmel-1 cells to produce IFN-$\gamma$ during the effector phase (7 days), it caused a reduction of IFN-$\gamma$-producing cells by 35 days (Fig. 1c), suggesting an impaired persistence of tumor-reactive T cells.

To test if impaired persistence of Ezh2-deficient T cells might be associated with their decreased capability to expand and elaborate effector function upon antigen rechallenge, we transferred equal numbers of WT and Ezh2$^{-/-}$ memory Pmel-1 cells, which were recovered from primary recipients 42 days after gp100-immunization, into secondary lymphopenic recipients. Seven days after gp100-rechallenge, Ezh2$^{-/-}$ memory Pmel-1 cells produced approximately 2- and 6-fold fewer total CD8$^+$ T cells and IFN-$\gamma$-secreting effectors, respectively, compared to their WT counterparts (Fig. 1d, e). In addition, when donor T cells derived from these secondary recipients were cultured with gp100 for 5 days, Ezh2$^{-/-}$ memory progenies were unable to expand compared to their WT counterparts (Fig. 1f). This suggests that decreased persistence of activated Ezh2$^{-/-}$ Pmel-1 cells may result from their impaired memory potential.

We used lymphoreplete mice to validate our conclusion. Equal number of congenically labeled WT (CD45.2$^+$, Thy1.1$^+$) and Ezh2$^{-/-}$ (CD45.2$^+$, Thy1.2$^+$) Pmel-1 cells were co-injected into non-irradiated B6/SJL mice (CD45.1$^+$, Thy1.2$^+$), followed by infection with vaccinia virus encoding gp100 (VVA-gp100). This also allowed us to test a cell-autonomous effect of Ezh2 deficiency. Again, although loss of Ezh2 did not impair the initial proliferation in the spleen 3 days after immunization, it caused approximately 2-fold reduction in number by 5 days and 35 days (Supplementary Fig. 2a). Impaired expansion of Ezh2$^{-/-}$ Pmel-1 cells was not the result of decreased proliferation but dependent on increased apoptosis (Supplementary Fig. 2b, c). Ezh2 deficiency decreased the frequency of Pmel-1 cells producing high levels of IFN-$\gamma$ 5 days and 35 days after immunization, without influencing IL-2 and granzyme B (GzmB) (Supplementary Fig. 2d). Ex vivo culture of 35 day-memory T cells revealed that Ezh2 deficiency resulted in 5-fold less expansion of Pmel-1 cells upon rechallenging with gp100 (Supplementary Fig. 2e). Furthermore, while gp100-restimulation induced a 2-fold higher frequency of IFN-$\gamma$-producing cells in WT 35 day-memory T cells compared to non-stimulation controls, it did not increase IFN-$\gamma$ production by Ezh2$^{-/-}$ memory T cells (Supplementary Fig. 2f). Thus, Ezh2 is crucial for persistence of effector and memory T cells under either lymphopenic or lymphoreplete conditions.

**Ezh2 promotes memory precursor formation.** A typical T-cell response contains three characteristic phases: clonal expansion, apoptotic contraction, and memory phase (Fig. 2a). We therefore used three different strategies to determine at which T-cell response phase(s) Ezh2 is required to support effector and memory T-cell persistence. During the initial phase, antigen-activated CD8$^+$ T cells generate two subsets of memory precursors: CD44$^{hi}$CD62L$^{hi}$ central memory precursor CD8$^+$ T cells (T$_{CMP}$) and CD44$^{hi}$CD62L$^{lo}$ T$_{EFF}$. T$_{CMP}$ are less differentiated and have greater capability than T$_{EFF}$ to replicate and generate memory cells[2,3,5–7]. We characterized T$_{CMP}$ and T$_{EFF}$ from WT and Ezh2$^{-/-}$ Pmel-1 cell recipients 4 days and 7 days after transfer into lymphodepleted mice. CD8$^+$ T cells expressing high levels of KLRG1 (KLRG1$^{hi}$) represent a terminally differentiated proliferating cells[3]. Ezh2 deficiency caused higher frequency of KLRG-1-expressing CD62L$^{hi}$- and CD62L$^{lo}$-T cells, which occurred 4 days after immunization and increased by 7 days (Fig.2b). This difference was even greater in PB (Fig. 2c). Ezh2 deficiency also induced a skewed differentiation of activated CD8$^+$ T cells towards T$_{EFF}$, as evidenced by increased fraction of

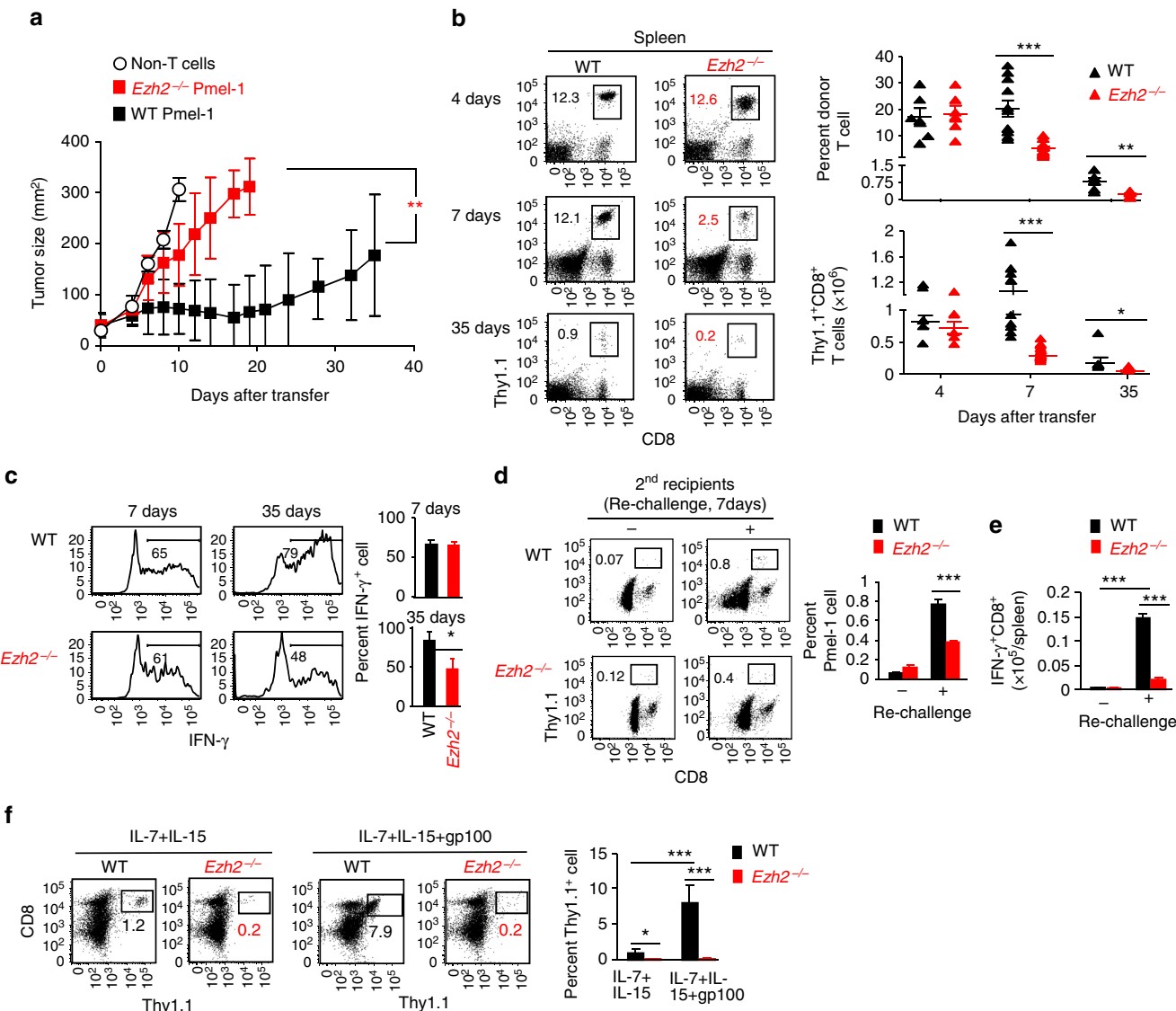

**Fig.1** Ezh2 is required for CD8[+] T cells to control tumor growth and promote memory precursor formation. **a** WT and $Ezh2^{-/-}$ naive Pmel-1 cells ($1 \times 10^6$, Thy1.1[+]) were transferred into sublethally irradiated (5 Gy) B6 mice (Thy1.2[+]) that had pre-established B16 melanoma, followed by treatment with IL-2 ($1 \times 10^5$ IU per injection, i.p., twice a day) and gp100-pulsed DCs (gp100-DCs, $1 \times 10^6$ per mouse, i.p.) for 3 days. Tumor size was monitored over time. **b-f** WT and $Ezh2^{-/-}$ $T_N$ Pmel-1 cells ($1 \times 10^6$, Thy1.1[+]) were transferred into sublethally irradiated non-tumor-bearing B6 mice, followed by immunization with IL-2 and gp100-DCs for 3d. **b** Donor T cells were collected from the spleen 4 days, 7 days, and 35 days after adoptive transfer. Plots and graphs show the frequency and numbers of donor T cells. **c** Percentage of IFN-γ-producing donor T cells in the spleen. **d** Donor T cells were collected from the spleen of WT and $Ezh2^{-/-}$ Pmel-1 cell primary recipients 42 days after transfer, and separately transferred into sublethally irradiated secondary non-tumor-bearing B6 mice ($4 \times 10^4$ cells per mouse), followed by treatment with IL-2 and gp100-DCs at 42 days, 43 days, and 44 days. By 49 days, donor T cells were collected from the spleen of the secondary mice. Plots and graph show the percentage of donor Pmel-1 cells. **e** Donor T cells derived from these secondary mice were activated with anti-CD3 Ab for 5 hrs to measure their production of IFN-γ. Graph shows the number of IFN-γ[+] Pmel-1 cells in the spleen. **f** Donor T cells collected at 49d from the secondary mice were cultured ex vivo with IL-7 + IL-15 in the presence or absence of gp100 for additional 5 days. Plots and graphs show the percentage of donor T cells in cultures. *$p < 0.05$, **$p < 0.01$, and ***$p < 0.001$ (two-tailed unpaired $t$ test). The data are representatives of three independent experiments with $n = 5$ mice per group in each (**a**; mean ± SD), or two experiments (**b-f**, $n = 3$–6 mice per group in each, mean ± SD)

$T_{EFF}$ and a corresponding reduction of both $T_{CMP}$ frequency and numbers throughout the expansion phase (Fig. 2d). Similar results were observed when responding cells were segregated into memory potential cells (MPC) and short-lived effector cells (SLECs) based on KLRG-1 and IL-7Rα expression, featured by increased percentage and number of SLECs at 4 days (Fig. 2d). Notably, while loss of Ezh2 did not affect the survival of $T_{CMP}$ at 4 days and 7 days, it caused enhanced apoptosis of $T_{EFF}$ early during expansion (4 days) (Fig. 2e). Since KLRG-1 normally is

not expressed on the surface of WT CD62L[hi]CD8[+] T cells[3,31], ectopic expression of KLRG-1 on the surface of Ezh2-deficient CD62L[hi]CD8[+] T cells probably indicates their precocious differentiation. Thus, Ezh2 is important for preserving the pool size of $T_{CMP}$, primarily through a mechanism of restraining differentiation into SLECs. This was supported by the observation that both $Ezh2^{-/-}$ $T_{CMP}$ and $T_{EFF}$ expressed 370- and 500-fold more transcripts, respectively, of the senescence gene $p19^{Arf}$ than their WT counterparts (Fig. 2f).

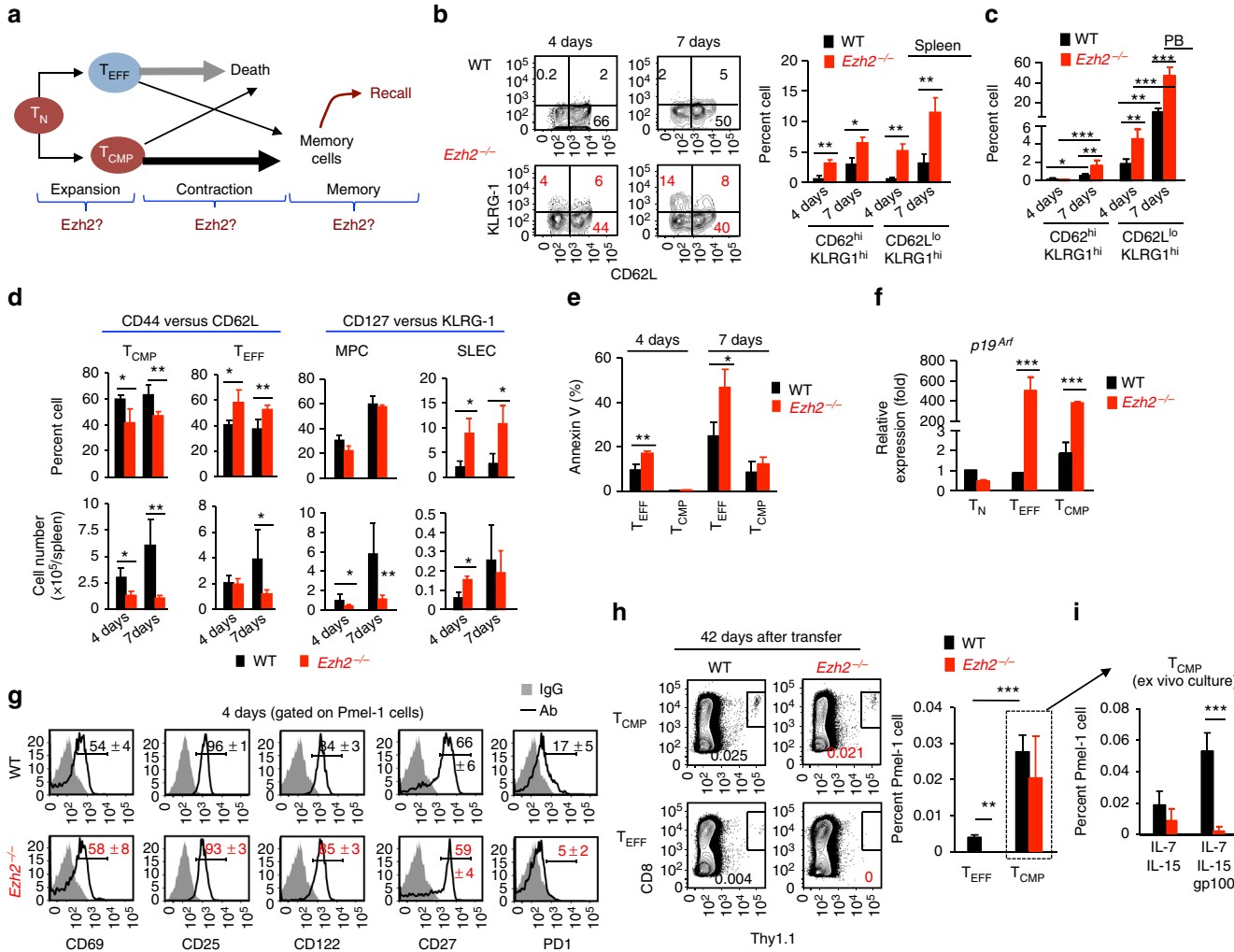

**Fig. 2** Ezh2 helps establish memory properties in activated CD8[+] T cells early during expansion. **a** Schematic diagram of three characteristic phases of the T-cell response and the possible role of Ezh2 in each phase. **b**–**g** WT and $Ezh2^{-/-}$ naive Pmel-1 cells (Thy1.1[+]) were transferred into sublethally irradiated non-tumor-bearing B6 mice (Thy1.2[+]), followed by immediate treatment with IL-2 and gp100-DCs for 3 days. Donor T cells were recovered at 4 days and 7 days after transfer. Plots and graphs show the percentage of KLRG-1-expressing (KLRG1[hi]) $T_{CMP}$ and $T_{EFF}$ in the spleen (**b**) and PB (**c**). **d** Graphs show the percentage and numbers of $T_{CMP}$ and $T_{EFF}$ (left panel) in the spleen at 4 days and 7 days after transfer. The right panel shows the percentage and numbers of MPCs and SLECs measured with KLRG-1 and CD127 at 4 days and 7 days after transfer. **e** The percentage of Annexin V-positive cells in the subpopulation of $T_{CMP}$ and $T_{EFF}$ at 4 days and 7 days after transfer. **f** Real-time RT-PCR measurement of $p19^{Arf}$ in the subset of $T_{CMP}$ and $T_{EFF}$ of 4 days and 7 days. **g** Histograms show the expression of indicated surface markers on WT and $Ezh2^{-/-}$ T cells derived from the spleen at 4d after transfer. **h**, **i** WT and $Ezh2^{-/-}$ naive Pmel-1 cells (Thy1.1[+]) were transferred into sublethally irradiated non-tumor-bearing B6 mice (Thy1.2[+]), followed by immediate treatment with IL-2 and gp100-DCs for 3 days. By 7 days after transfer, donor $T_{CMP}$ and $T_{EFF}$ were highly purified using FACS sorter, and transferred into sublethally irradiated secondary recipients that had been immunized with gp100-DCs 7 days earlier (described in Supplementary Fig.4). Forty-two days later, donor T cells were collected from the spleen of these secondary recipients (**h**), and further cultured ex vivo for additional 5 days (**i**). Plots and graphs show the frequency of donor T cells derived from the secondary recipients of WT and $Ezh2^{-/-}$ $T_{CMP}$ and $T_{EFF}$. *$p < 0.05$, **$p < 0.01$, and ***$p < 0.01$ (two-tailed unpaired $t$ test). The data are representative of four independent experiments with $n = 3$ mice per group in each (**b**–**g**; mean ± SD) or two experiments with $n = 4$ mice per group in each (**h**, **i**, mean ± SD)

Ezh2-deficiency-mediated change in surface phenotype was largely associated with T-cell differentiation rather than activation and exhaustion markers (e.g., CD69, CD25, CD122, and PD-1) in these lymphodepleted hosts (Fig. 2g). Using lymphoreplete mice immunized by VVA-gp100, we validated the impact of Ezh2 deficiency on reducing $T_{CMP}$ and MPCs and increasing SLECs without changing the expression of CD122 and PD-1 during expansion phase (Supplementary Fig. 3). To examine the cell autonomous effect of Ezh2 on endogenous CD8[+] T-cell responses, we co-transferred equal amount of WT (Thy1.1[+]CD45.2[+]) and $Ezh2^{-/-}$ (Thy1.1[−]CD45.2[+]) splenocytes into lymphoreplete B6/SJL mice (Thy1.1[−]CD45.1[+]), followed by infection with vaccinia virus encoding OVA (VVA-OVA). As compared to WT CD8[+] T cells, $Ezh2^{-/-}$ CD8[+] T cells produced 1.5- to 2-fold more OVA$_{257-264}$-specific T cells 3 days after infection, maintained at 4 days, and dramatically declined by 5 days (Supplementary Fig. 4a, b). Notably, the increase of OVA$_{257-264}$-specific $Ezh2^{-/-}$ CD8 T cells 3 days after infection was associated with enhanced proliferation rates (Supplementary Fig. 4c), increases of KLRG1[hi] cells (Supplementary Fig. 4d, e) and decreased ratio of $T_{CMP}$ vs. $T_{EFF}$ (Supplementary Fig. 4f). Thus, loss of Ezh2 leads to an earlier occurrence of T-cell response peak, exaggerated terminal differentiation and loss of memory precursors.

The second phase is featured by massive apoptosis of effector cells and the transition of memory precursors into mature memory cells. To evaluate if Ezh2 deficiency affected this transition, we purified $T_{CMP}$ and $T_{EFF}$ from primary lymphodepleted recipients 7 days after transfer of Pmel-1 cells and immunization, and separately transferred them into gp100 immunization-matched secondary recipients (Supplementary Fig. 5). WT $T_{CMP}$ produced 6-fold more memory cells than WT $T_{EFF}$ 42 days after transfer (Fig. 2h), confirming that $T_{CMP}$ have greater ability than $T_{EFF}$ to produce memory cells[3,32]. $Ezh2^{-/-}$ $T_{EFF}$ failed to produce detectable memory T cells. However, $Ezh2^{-/-}$ $T_{CMP}$ produced similar percentages of memory-phenotype cells as did WT $T_{CMP}$ (Fig. 2h), but their progenies were unable to expand upon gp100 rechallenge, unlike those of WT $T_{CMP}$ (Fig. 2i). Thus, Ezh2 is dispensable for the homeostatic survival of $T_{CMP}$ during contraction phase, but is important for the transition of both $T_{CMP}$ and $T_{EFF}$ into mature memory T cells.

Increased apoptosis of $Ezh2^{-/-}$ T cells during the effector phase might result in long-term consequences on recall response capacity during the final memory phase. To test this, we immunized lymphoreplete $Ezh2^{fl/fl}$ mice with VVA-gp100 (Supplementary Fig. 6a). Forty-two days later when gp100-specific memory $CD8^+$ T cells were formed (Supplementary Fig. 6b), we isolated splenocytes from these mice and treated them with TAT-Cre to delete Ezh2 (Supplementary Fig. 6c), followed by culturing them ex vivo for 5 days, with or without gp100 addition. Deletion of Ezh2 by TAT-Cre dramatically decreased the capacity of these $Ezh2^{fl/fl}$ memory $CD8^+$ T cells to expand and produce IFN-γ upon gp100 rechallenge (Supplementary Fig. 6d, e). These results indicate that Ezh2 is important for maintaining the recall response capacity of developed memory T cells.

**Ezh2 orchestrates gene programs of memory properties.** To identify the Ezh2-targeted genes associated with effector and memory differentiation, we performed RNA sequencing of WT and $Ezh2^{-/-}$ Pmel-1 cells after TCR activation for 3 days. Ezh2 deficiency had minimal effect on gene expression in $T_N$ (Fig. 3a; Supplementary Data 1). In contrast, TCR activation of $Ezh2^{-/-}$ Pmel-1 cells led to the up-regulation of 279 genes and down-regulation of 168 genes compared to their WT counterparts (Fig. 3a; Supplementary Data 2). Genes altered in activated $Ezh2^{-/-}$ Pmel-1 cells were associated with cellular proliferation, cell death and survival, as well as cell function and maintenance (Fig. 3b). Ezh2 deficiency upregulated Id2, Prdm1, and Eomes (Fig. 3c), all critical for effector differentiation and functionality[1,3], and decreased Id3, which controls memory formation[33–35]. RT-PCR analysis validated these changes (Fig. 3d, e). RNA-seq gene profiling confirmed the role of Ezh2 in restraining effector differentiation, as evidenced by upregulated effector molecules and chemokine receptors in TCR-activated $Ezh2^{-/-}$ $CD8^+$ T cells (Supplementary Fig. 7a).

To assess the impact of Ezh2 deficiency on TF expression in memory precursors, we isolated $T_{CMP}$ and $T_{EFF}$ from WT and $Ezh2^{-/-}$ Pmel-1 cell recipients 4 days after activation. As compared to WT $T_{CMP}$, $Ezh2^{-/-}$ $T_{CMP}$ had lower levels of Id3, but higher expression of Id2 and Eomes (Supplementary Fig. 7b). In $T_{EFF}$, Ezh2 deficiency led to increased expression of Id2, Eomes, and Prdm1 (Supplementary Fig. 7b). We performed chromatin immunoprecipitation (ChIP) analysis and observed that Ezh2 bound to the promoter regions of these gene loci (Supplementary Fig. 7c). This was confirmed using chromatin from activated $Ezh2^{-/-}$ Pmel-1 cells 3 days after activation as evidenced by decreased amount of Ezh2 and H3K27me3 at the promoter region of these gene loci (Supplementary Fig. 7d). Ezh2

appeared to have differential effects on Tbx21 (which encodes T-bet) expression between $T_{CMP}$ vs. $T_{EFF}$ via an unknown mechanism (Supplementary Fig. 7b, c). Since $CD8^+$ T cells that express high levels of Prdm1, Id2 and Tbx21 but low levels of Id3 are reported to undergo accelerated and enhanced terminal differentiation;[1,3,33,34] we propose that Ezh2 orchestrates the expression of these TFs for controlling stepwise effector differentiation and memory formation of $CD8^+$ T cells.

**Phosphorylation of Ezh2 impairs the maintenance of memory precursors.** During the immune response, normal $CD8^+$ T cells express high levels of Ezh2 upon antigen activation, however, they still undergo a "programmed" differentiation into terminal $T_{EFF}$. This points to a mechanism that modifies Ezh2 function in T cells, reducing Ezh2 regulation of memory T cells. We found that TCR-activated Pmel-1 cells expressed 42- and 23-fold higher Ezh2 protein 3 days and 7 days after culture, respectively, than $T_N$ (Fig. 4a). However, as compared to $T_N$, activated $CD8^+$ T cells showed a profound decrease in Ezh2 function, as evidenced by the fact that 3 day- and 7 day-Pmel-1 cells contained 2- and 5-fold less H3K27me3, respectively (Fig. 4b), increase of Ezh2-silenced genes Id2, Eomes and Prdm1 while decreasing Ezh2-activated gene Id3 (Fig. 4c), and impaired antitumor activity (Fig. 4d). ChIP analysis showed that in $CD8^+$ $T_N$, Ezh2 bound to the promoter region of Id3, Id2, Prdm1, and Eomes (Fig. 4e). However, the amount of Ezh2 within these promoter regions was dramatically decreased in proliferating $CD8^+$ T cells, which occurred 3 days after activation and persisted throughout 7 days (Fig. 4f). This decreased presence of Ezh2 was paralleled by a reduction of H3K27me3 at the Prdm1 and Eomes loci (Fig. 4g). Thus, Ezh2 is dissociated from the promoter regions of these TFs as early as 3 days after activation.

To determine if $T_{CMP}$ and $T_{EFF}$ differentially modify Ezh2 function, we recovered them from WT Pmel-1-cell recipients 7 days after activation. As compared to $T_N$, $T_{EFF}$ expressed higher levels of Ezh2 (Supplementary Fig. 8a), upregulated Id2, Prdm1 and Eomes, and decreased Id3 (Supplementary Fig. 8b), and reduced binding of Ezh2 at the regulatory regions of Id3, Prdm1 and Eomes (Supplementary Fig. 8c). $T_{CMP}$ decreased the amount of Ezh2 within the promoters of Id3, Eomes and Prdm1 compared to $T_N$, but they retained more Ezh2 at the promoter regions of these gene loci than $T_{EFF}$ (Supplementary Fig. 8c). This correlated with higher levels of Id3 transcripts but lower transcription of Prdm1 and Eomes in $T_{CMP}$ than $T_{EFF}$ (Supplementary Fig. 8b). Thus, $T_{EFF}$ have more dramatic reduction of Ezh2 function compared to $T_{CMP}$.

Phosphatidylinositol-3-kinase (PI3K)/Akt is important for T-cell proliferation and differentiation[13]. In cancer cells, Akt phosphorylates Ezh2 at serine 21 ($pEzh2_{S21}$), thereby suppressing Ezh2 enzymatic activity[36]. We assessed if Akt phosphorylation of Ezh2 might induce the dissociation of Ezh2 from these TF loci in proliferating T cells. As compared to $T_N$, $T_{CMP}$ and $T_{EFF}$ from B6 mice that received Pmel-1 cells 7 days earlier had 8.3-fold and 3.7-fold more $pEzh2_{S21}$, respectively (Supplementary Fig. 8d). This correlated with decreased H3K27me3 and increased pAkt (Supplementary Fig. 8a, d). $pEzh2_{S21}$ occurred in TCR-activated $CD8^+$ T cells 3 days after culture. By 5 days and 7 days, activated cells expressed 2.3-fold and 8.4-fold more $pEzh2_{S21}$ than $T_N$ (Fig. 5a). Treatment with MK2206, an allosteric inhibitor of Akt, led to a marked decrease of $pEzh2_{S21}$ and increase of H3K27me3 in proliferating $CD8^+$ T cells (Fig. 5b–d). Inhibiting Akt or its upstream activator PI3K by PI103 likewise upregulated Id3 expression while reducing Id2, Eomes and Prdm1 (Fig. 5e), affirming an orchestrated regulatory pathway across these TFs. Inhibiting Akt activity also increased the amount of Ezh2 and

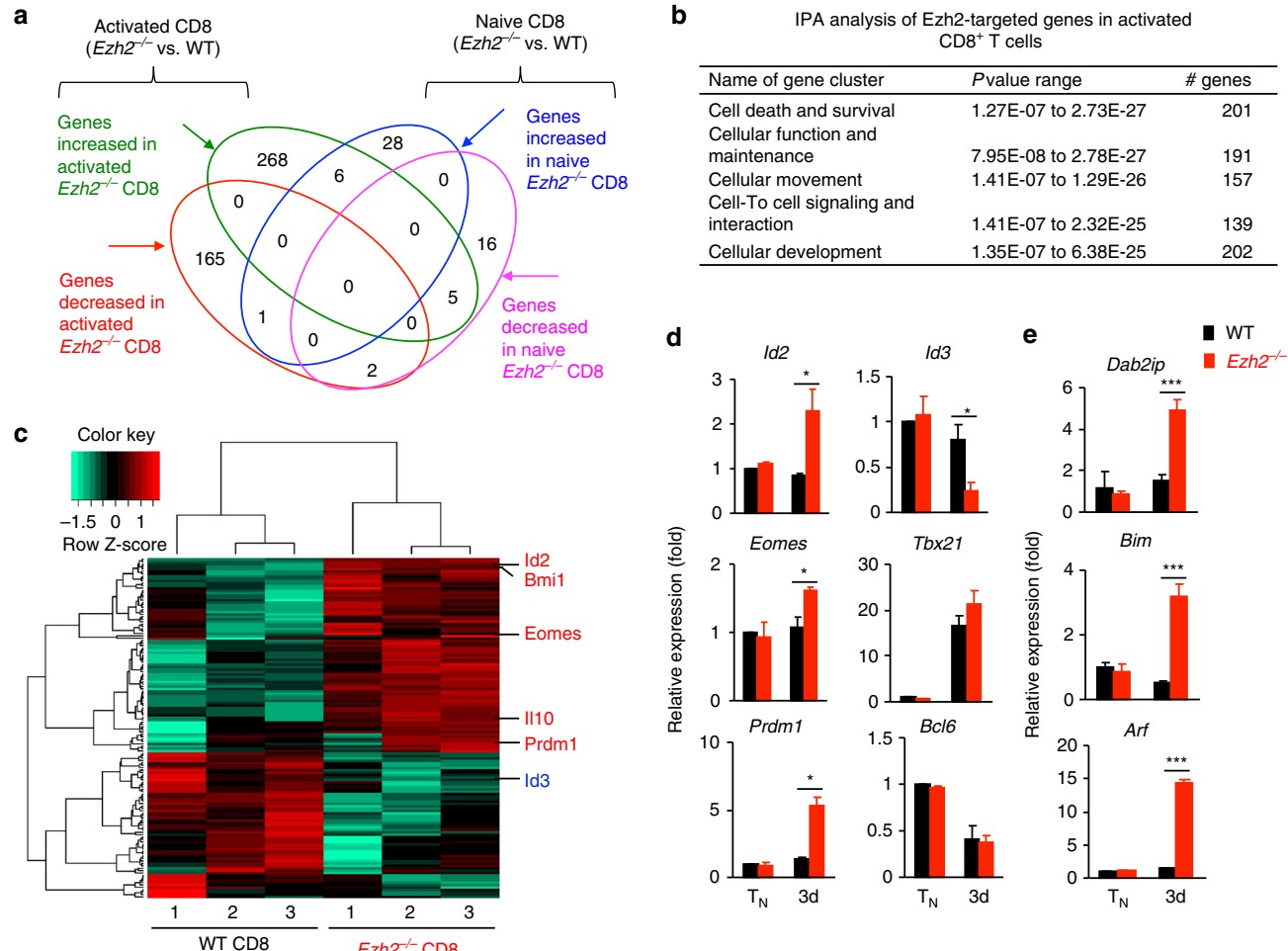

**Fig.3** Ezh2 orchestrates the expression of genes critical for effector differentiation and memory formation. **a–c** WT and *Ezh2*[−/−] Pmel-1 cells were cultured in the presence of anti-CD3/CD28 Abs and IL-2 for 3 days. Cells were collected to extract RNA for sequencing. Using one-way ANOVA analysis, we selected transcripts with $p < 0.01$ and $q < 0.01$ for comparing paired groups and at least a 1.5-fold difference from the means for the paired groups. **a** Venn diagram shows the number of differentially expressed genes. **b** Table shows the signaling pathways most regulated by Ezh2 in activated Pmel-1 cells, which was identified by INGENUITY pathway analysis. **c** Heat map shows the cluster of genes associated with Cell Function and Maintenance in activated WT and *Ezh2*[−/−] Pmel-1 cells. **d**, **e** Gene expression was assessed by real-time RT-PCR. *$p < 0.05$, and ***$p < 0.001$ (two-tailed unpaired *t* test). The data are representative of four independent experiments (**d**, **e**; mean ± SD)

H3K27me3 at the promoter regions of *Id3*, *Id2*, *Eomes*, and *Prdm1* loci (Fig. 5f, g). Rapamycin inhibition of mTOR, a downstream effector of Akt pathway[13,37], increased *Eomes* expression and reduced *Id3* (Fig. 5e), suggesting that the mechanism of Ezh2 inhibition by Akt differs from mTOR-dependent effects.

To establish a specific effect of Akt-mediated phosphorylation on Ezh2, we infected Pmel-1 cells with retrovirus encoding Akt-phosphorylation-resistant Ezh2 in which the serine at amino acid 21 was replaced by alanine (named Ezh2$_{S21A}$). Pmel-1 cells expressing Ezh2$_{S21A}$ increased H3K27me3 compared to Ezh2 control (Fig. 5h), upregulated *Id3* transcript and repressed *Eomes* and *Prdm1* expression (Fig. 5i), and increased Ezh2 binding at *Id3*, *Id2*, *Eomes* and *Prdm1* loci (Fig. 5j). Notably, introduction of Ezh2$_{S21A}$ increased the H3K27me3 level within the *Eomes* and *Prdm1* loci but not the *Id3* and *Id2* loci (Fig. 5k), suggesting that epigenetic regulation of these loci occurs via a different mechanism. Using lymporeplete hosts infected by VVA-gp100, we confirmed that WT Pmel-1 cells induced Ezh2 and pEzh2$_{S21}$ 3 days after activation and markedly increased by 5 days (Supplementary Fig. 9a, b), which was inversely correlated to the decreased cellular H3K27me3 (Supplementary Fig. 9c). In

aggregate, Akt activation profoundly reduces Ezh2 function in activated CD8[+] T cells.

To define the impact of Ezh2 phosphorylation by Akt in CD8[+] T cells, we made a retrovirus encoding a phosphomimetic Ezh2, in which serine 21 was replaced by aspartate (named Ezh2$_{S21D}$)[36]. We separately introduced Ezh2$_{S21A}$ and Ezh2$_{S21D}$ into activated *Ezh2*[−/−] Pmel-1 cells to assess their specific effects in T cells without endogenous Ezh2 (Fig. 6a). As compared to Ezh2$_{S21D}$, Ezh2$_{S21A}$ increased H3K27me3 in activated CD8[+] T cells (Fig. 6b), and induced high levels of Id3 but reduced Eomes and Prdm1 transcripts in these T cells 7 days after activation (Fig. 6c). Upon transfer into lymphopenic B6 mice and treatment with IL-2 and gp100-DCs, *Ezh2*[−/−] Pmel-1 cells overexpressing Ezh2$_{S21A}$ produced more donor T cells (Figs. 6d) and 6-fold more Id3[hi] donor cells, compared to Ezh2$_{S21D}$ 6d after transfer (Fig. 6e). Overexpressing Ezh2$_{S21A}$ caused 3-fold smaller in frequency of KLRG1[hi] T cells than Ezh2$_{S21D}$, without changing the fraction of CD62L[hi] cells and IFN-γ-producing cells (Fig. 6e). As compared to Ezh2$_{S21A}$, overexpressing WT Ezh2 in *Ezh2*[−/−] Pmel-1 cells was able to rescue their survival in vivo (Fig.6d), but less effective in modifying the expression of *Id3*, *Prdm1* and *Eomes* (Fig.6c) and unable to sustain Id3[hi] T cells (Fig.6e). These investigations

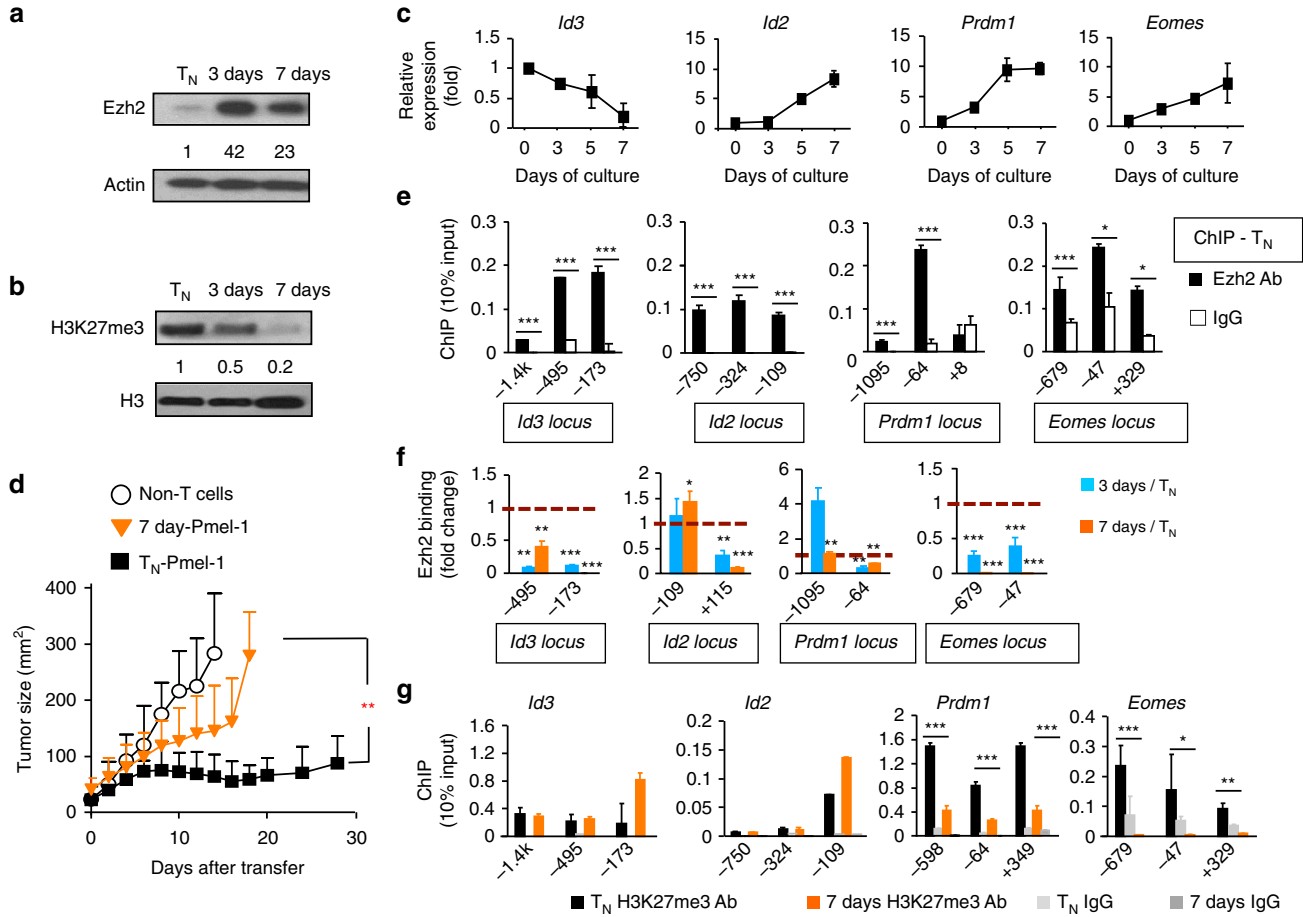

**Fig. 4** Ezh2 is dissociated from the regulatory regions of key TFs during CD8[+] T-cell expansion. WT naive Pmel-1 cells were stimulated with anti-CD3/CD28 Ab + IL-2 for 7 days. Cells were collected at 0 days, 3 days and 7 days. **a**, **b** Immunoblot analysis of Pmel-1 cells before and after TCR-activation, probed with anti-Ezh2 Ab (**a**) and anti-H3K27me3 Ab (**b**). **c** Real-time RT-PCR analysis of gene expression in Pmel-1 cells before and after activation at indicated time points. **d** Tumor size in B16 tumor-bearing B6 mice receiving no transfer (Non-T cells), transfer of WT naive Pmel-1 cells ($T_N$) or in vitro TCR-activated 7 days Pmel-1 cells and treatment with IL-2 and gp100-DCs for 3 days. **e**–**g** ChIP analysis of $T_N$ (**e**), 3 days and 7 days Pmel-1 cells (**f**), and $T_N$ and 7 days Pmel-1 cells (**g**). *$p < 0.05$, **$p < 0.01$, and ***$p < 0.001$ (two-tailed unpaired t test). The data are representative of four independent experiments (**a**–**c**), two experiments with $n = 5$ mice per group in each (**d**; mean ± SD), or three experiments (**e**–**g**; mean ± SD)

identify Akt-mediated phosphorylation of Ezh2 as a novel and important mechanism regulating effector differentiation and memory formation, beyond T-cell survival.

**Akt-insensitive Ezh2 augments antigen-specific T-cell antitumor efficacy**. ACT for cancer requires sufficient amplification and persistence of tumor-specific T cells to eradicate the tumor[4,6,7,11]. Current in vitro methods to expand cells to sufficient numbers impair the maintenance of memory properties in cultured T cells[6,7]. Ex vivo treatment of expanding CD8[+] T cells with an Akt inhibitor led to increase of memory-phenotype cells and antitumor immunity[38,39]. Indeed, transfer of Pmel-1 cells treated by MK2206 during ex vivo culture induced greater antitumor activity than untreated Pmel-1 cells, but was less potent than Ezh2$_{S21A}$-transduced Pmel-1 cells (Fig. 7a). As compared to MK2206-treated Pmel-1 cell recipients, Ezh2$_{S21A}$-Pmel-1 cell recipients generated higher frequency of total Pmel-1 cells and Id3$^{hi}$ Pmel-1 cells, but a lower frequency of KLRG1$^{hi}$ cells in the spleen 9 days after transfer (Fig. 7b).

Enhanced tumor immunity by Ezh2$_{S21A}$-Pmel-1 cells could potentially result from increased expression of overall Ezh2 protein. To test it, we assessed whether overexpressing normal

Ezh2 in Pmel-1 cells, which is susceptible to Akt phosphorylation, may influence their antitumor activity. Transfer of Pmel-1 cells expressing Ezh2$_{S21A}$ had dramatically enhanced capacity to inhibit tumor growth compared to either Ezh2 or GFP control (Fig. 7c). Ezh2$_{S21A}$-, Ezh2- and GFP-Pmel-1 cell recipients of B16 melanoma had similar amounts of donor T cells in PB within 7d after transfer. Over time, Ezh2$_{S21A}$-Pmel-1 cell recipients produced approximately 10-fold more in frequency of donor T cells than either Ezh2- or GFP-Pmel-1 cell recipients between 10 and 14 days (Fig. 7d). Upon rechallenge with gp100-DCs 18 days after transfer, Ezh2$_{S21A}$-Pmel-1 cell recipients had ~4-fold more circulating Pmel-1 cells than Ezh2- and GFP-Pmel-1 cell recipients by 24 days (Fig. 7d), demonstrating an enhanced recall response. We also found that ectopic expression of Ezh2$_{S21A}$ induced a lower frequency of KLRG1$^{hi}$ cells in vivo 10 days after transfer compared to Ezh2 and GFP controls (Fig. 7e). Analysis of tumor infiltrating lymphocytes (TILs) showed that as compared to GFP- and Ezh2-Pmel-1 cell recipients, Ezh2$_{S21A}$-Pmel-1 cell recipients had 1.5-fold higher frequency of TILs (Figs. 7f), 5-fold more total GFP[+] TILs and 4-fold more IFN-γ-producing GFP[+] TILs (Fig. 7g). Overexpression of Ezh2$_{S21A}$ induced 2-fold and 4-fold more CD62L$^{hi}$ CD8[+] T cells in the spleen and tumor, respectively, than GFP- and Ezh2-Pmel-1 cell recipients, but did

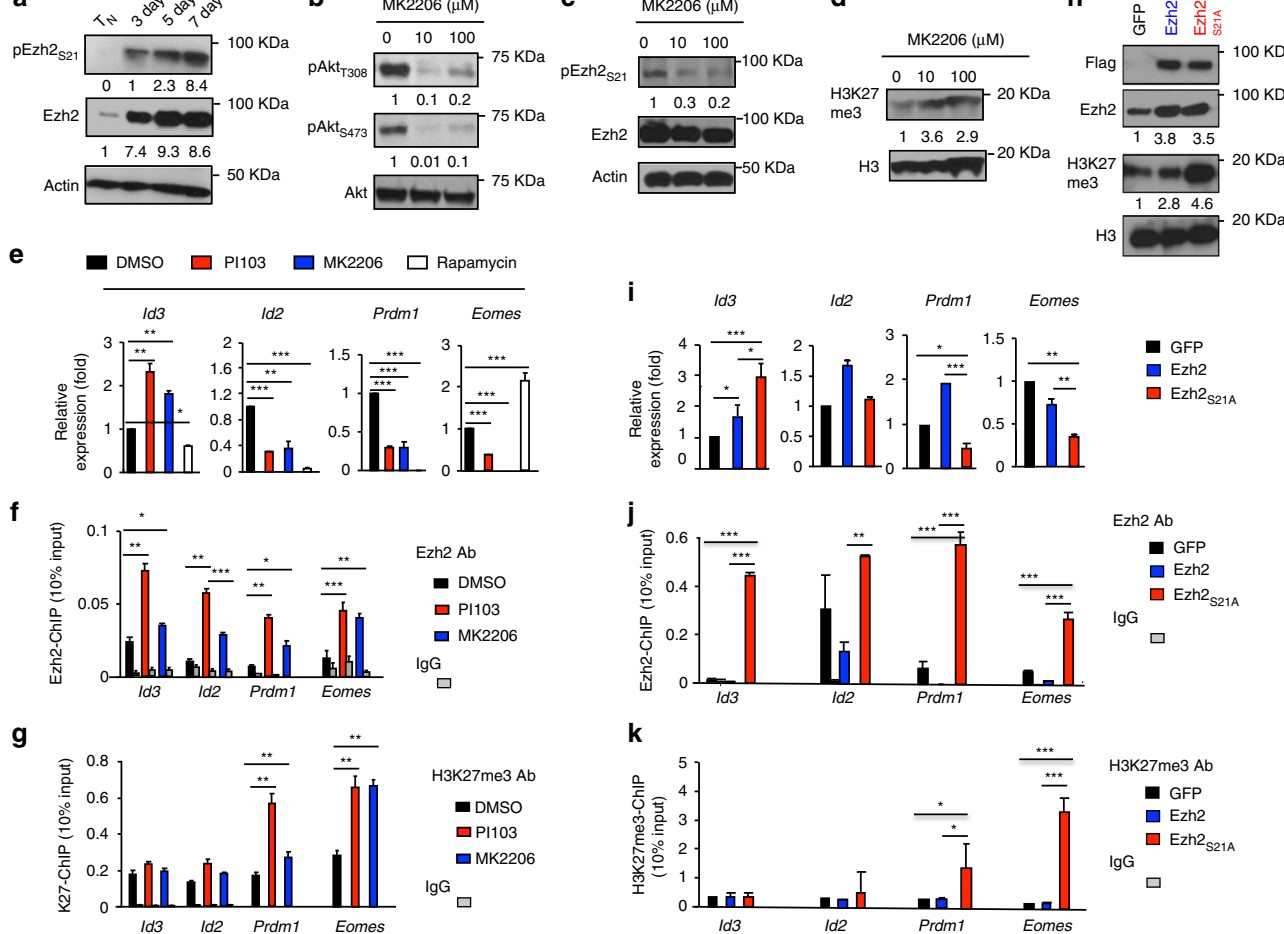

**Fig. 5** Phosphorylation of Ezh2 by Akt dissociates Ezh2 from the promoter regions of major TF loci. **a** WT Pmel-1 cells were stimulated with anti-CD3/CD28 Ab + IL-2. Immunoblot analysis of Pmel-1 cells stimulated in vitro for 3 days, 5 days, and 7 days, probed with Abs against Ezh2 and phosphorylated Ezh2_{S21}. Unstimulated $T_N$ were used as control. **b–d** Immunoblot analysis of Pmel-1 CD8+ T cells stimulated in vitro for 7 days, with or without treatment of MK2206, probed with indicated Abs. **e–g** WT Pmel-1 cells were cultured with anti-CD3/CD28 Ab + IL-2, with or without treatment of PI103, MK2206 or rapamycin for 7 days. Real-time RT-PCR analysis of Ezh2-targeted genes (**e**). ChIP analysis of cultured Pmel-1 cells treated with PI103, or MK2206. Graphs show the deposition of Ezh2 (**f**) and H3K27me3 (**g**) at the promoter regions of Id3, Id2, Prdm1, and Eomes. **h–k** WT Pmel-1 cells were stimulated with anti-CD3/CD28 Ab for 36 h, followed by infection with MigR1 retrovirus (GFP) or MigR1 retrovirus encoding flag-tagged Ezh2, or Ezh2_{S21A}. Pmel-1 cells were collected at 7 days after culture. Immunoblot analysis of GFP and Ezh2_{S21A} Pmel-1 cells, probed with indicated Abs (**h**). Real-time RT-PCR analysis of their expression of major TFs (**i**). ChIP analysis shows the deposition of Ezh2 (**j**) and H3K27me3 (**k**) at the promoter regions of these major TFs loci. *$p < 0.05$, **$p < 0.01$, and ***$p < 0.001$ (two-tailed unpaired $t$ test). The data are representatives of three independent experiments (**a–d**), or two experiments (**e–k**; mean ± SD)

not prevent Pmel-1 cells from differentiating into CD62L^lo $T_{EFF}$ in vivo (Fig. 7h). Thus, overexpressing Ezh2_{S21A} dramatically improves the persistence of tumor-reactive CD8+ T cells in vivo upon chronic exposure to tumor antigen.

**Critical role of Id3 in Ezh2-regulated T-cell recall response.** Id3 deficiency leads to impaired recall response capability of antigen-driven CD8+ memory T cells[33,34], resembling Ezh2 inhibition. To understand whether Id3 mediates Ezh2-regulated recall response, we retrovirally introduced Id3 into Ezh2^{−/−} Pmel-1 cells (Supplementary Fig. 10). Adoptive transfer experiments showed that as compared to control GFP-transduction, overexpressing Id3 improved the survival and persistence of Ezh2^{−/−} Pmel-1 cells in vivo 35d after immunization (Fig. 8a), decreased the fraction of KLRG1^hi cells (Fig. 8b) and restored IFN-γ production of these memory Ezh2^{−/−} Pmel-1 cells upon gp100 rechallenging ex vivo (Fig. 8c). Id3-overexpression rescued memory Ezh2^{−/−} Pmel-1 cells to a comparable level of Ezh2-overexpression (Fig. 8a-c).

However, neither Id3- nor Ezh2-overexpression completely restored the capability of Ezh2^{−/−} Pmel-1 cells to produce memory cells like GFP-transduced WT Pmel-1 controls (Fig. 8a–c). Since Ezh2 was critical for generating memory precursors within 5 days of activation (Fig. 2 and Supplementary Fig. 3), and since retrovirally introduced genes did not reach peak expression by 5 days after T-cell activation[23], one plausible explanation for the partial rescue effect of Id3 is its delayed expression in Ezh2^{−/−} Pmel-1 cells.

To assess the direct impact of Ezh2 on Id3 activation, we cloned the promoter region of the Id3-gene (ranging from −1.5 kb upstream of the transcription start site (TSS) to + 0.5Kb downstream of the TSS) into the pGL3 luciferase reporter vector and constructed an Id3-specific pGL3 reporter (Id3-pGL). Overexpressing Ezh2_{S21A}, but not Ezh2_{S21D}, activated Id3 transcription (Fig. 8d). To evaluate the effect of Ezh2 enzymatic activity on Id3 transcription, we deleted endogenous Ezh2 in 3T3 cells using CRISPR (Fig. 8e), followed by viral transduction of various Ezh2 mutants. Mutant Ezh2_{S21A} had a greater capacity than Ezh2 to

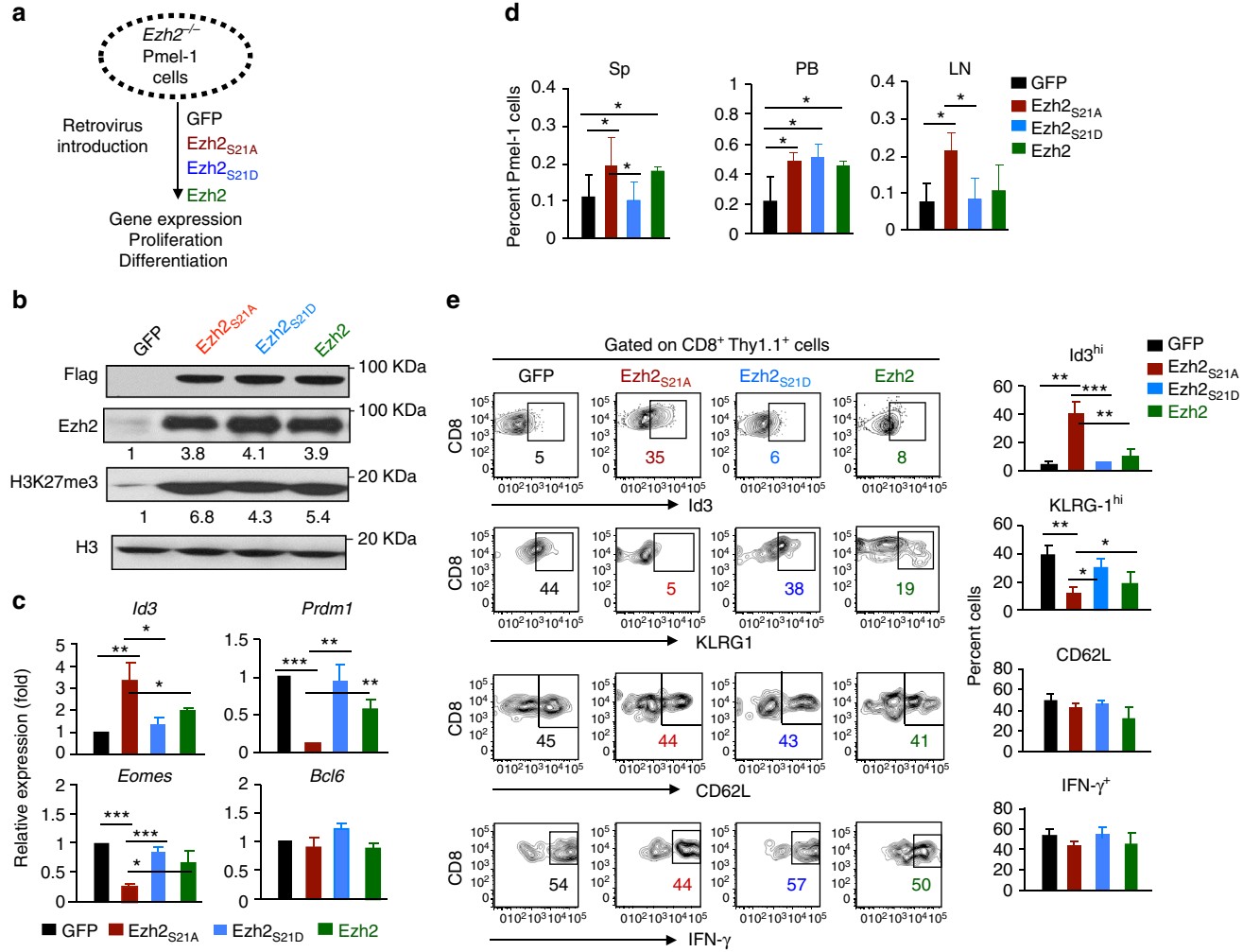

**Fig. 6** Inhibiting Akt-mediated phosphorylation of Ezh2 enhances the generation of $T_{CMP}$. $Ezh2^{-/-}$ Pmel-1 cells (Thy1.1$^+$) were stimulated with anti-CD3/CD28 Ab + IL-2 for 36 h, followed by infection with MigR1 retrovirus encoding GFP, Ezh2, Ezh2$_{S21D}$ and Ezh2$_{S21A}$, respectively. At 7 days, after stimulation, these infected T cells were collected. **a** Experiment scheme. **b** Immunoblot analysis of Pmel-1 cells transduced with GFP, Ezh2, Ezh2$_{S21D}$ and Ezh2$_{S21A}$, probed with Flag, H3K27me3 and H3. **c** RT-PCR analysis of gene expression in Pmel-1 cells transduced by indicated genes. **d, e** These 7 day-Pmel-1 cells (Thy1.1$^+$) were transferred into sublethally irradiated B6 mice (Thy1.2$^+$), followed by treatment with IL-2 and gp100-DCs for 3d after transfer. **d** Donor T cells were collected from the spleen, PB and LN at 6d after adoptive transfer, and measured by flow cytometric analysis. **e** Plots and graphs show the percentage of of Id3$^{hi}$, KLRG1$^{hi}$, CD62L$^+$ and IFN-γ$^+$ within donor T cells from the spleen. *$p < 0.05$, **$p < 0.01$, and ***$p < 0.001$ (two-tailed unpaired $t$ test). The data are representatives of three experiments (**b, c**; mean ± SD) or two experiments with $n = 4$ mice per group in each (**d, e**; mean ± SD)

activate *Id3*, whereas the loss-of-function mutant Ezh2$_{H689A}$[40] failed to activate *Id3* (Fig. 8f), confirming that Ezh2 requires its methyltranferase activity to activate *Id3* transcription.

To understand the mechanism by which Ezh2 stimulated Id3 transcription, we assessed if Ezh2 influenced histone modification states at the promoter region of Id3. Recent studies demonstrate that TFs (e.g., Tbx21, Prdm1, Eomes and Irf4) are bivalent for both H3K27me3$^+$ and H3K4me3$^+$ histone methylation under naive state[18]. We observed that the Id3 locus was marked by both H3K27me3 and H3K4me3 in CD8$^+$ T$_N$ (Fig. 8g). TCR-activated CD8$^+$ T cells reduced H3K4me3 and Pol II at Id3 locus upon differentiation (Fig. 8g). Loss of Ezh2 led to decreased deposition of H3K4me3 and Pol II at Id3 locus, but not Id2 locus (Fig. 8h). Thus, Ezh2 activation of Id3 transcription in T cells likely involves the enzyme(s) that catalyzes H3K4me3. Akt phosphorylation of Ezh2 leads to dissociation of Ezh2 from the Id3 locus and subsequent resolution into non-permissive state. This is in contrast to a recent study showing that in cancer cells, Akt-induced pEzh2$_{S21}$ activates those genes lacking H3K27me3 modifications[24].

## Discussion

Understanding the epigenetic mechanisms that mediate continual replication of antigen-driven T cells without senescence during reaction to chronic infection and cancer may lead to new strategies to improve the efficacy of ACT[4,7,8,11,32,41–43]. Minimally differentiated CD8$^+$ T cells, including T$_N$ and T$_{CMP}$, exhibit superior persistence and enhanced antitumor activity compared with differentiated T$_{EFF}$[4,6,10,44]. The goal therefore is to generate cellular products rich in less-differentiated memory precursors, however, the mechanisms that restrain effector differentiation and maintain memory potential remains poorly understood. T-cell signal strength, which determines the quantity and quality of T-cell responses and includes signals from the TCR, costimulatory molecules and cytokines, is believed to be epigenetically controlled. However, the epigenetic regulator(s), responsible for converting these signals into the transcriptional programs that generate and maintain memory precursors, has not been fully identified[2,45–50]. Thus, our observations that Ezh2 regulates effector, memory precursor and antigen-specific recall response have profound implications towards the development of new

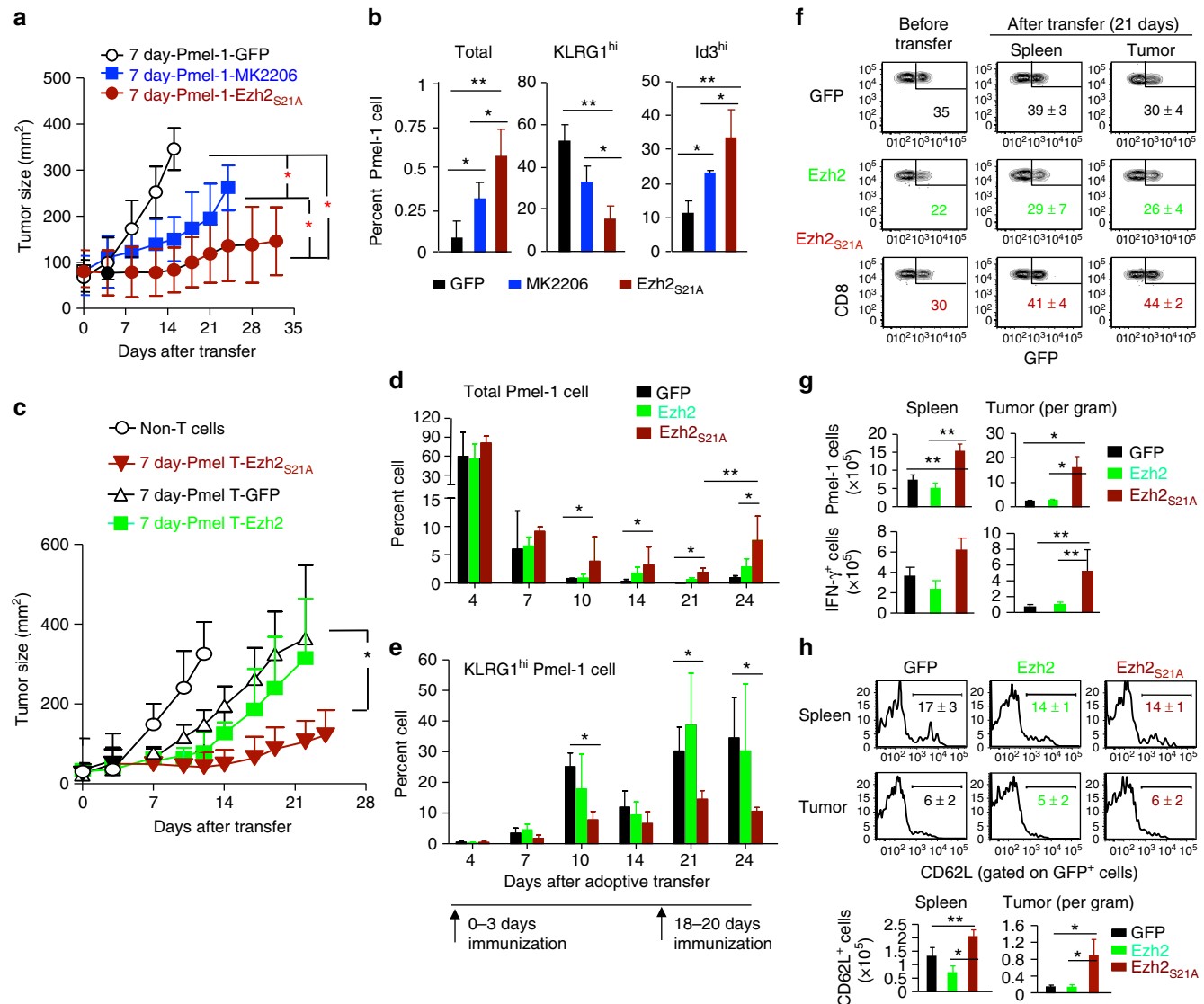

**Fig. 7** Inhibiting Akt-mediated Ezh2 phosphorylation improves antitumor immunity. **a**, **b** Ex vivo-expanded 7 day-Pmel-1 cells (Thy1.1+), MK2206-treated 7 day-Pmel-1 cells (Thy1.1+), or 7 day-Pmel-1 cells infected with MigR1 retrovirus encoding Ezh2$_{S21A}$ (Thy1.1+, $5 \times 10^5$ T cells per mouse) were transferred into sublethally irradiated B6 mice that had pre-established B16 melanoma, followed by treatment with IL-2 and gp100-DCs from 0 to 3 days and repeated once more from 18 to 20 days. Tumor size was monitored over time (**a**). Some recipient mice were sacrificed at 9 days to examine the presence of donor Pmel-1 cells in the spleen using flow cytometric analysis. Graphs show the percentage of indicated total cells and cell subsets (**b**). **c–h** B6 mice with pre-established B16 melanoma cells received no adoptive transfer (Non-T cells), transfer of in vitro expanded 7 day-Pmel-1 cells infected with MigR1 encoding GFP, Ezh2 or Ezh2$_{S21A}$ ($5 \times 10^5$ T cells per mouse), followed by treatment with IL-2 and gp100-DCs from 0-3 days and repeated once more from 18 to 20 days. Tumor size was monitored over time (**c**). The percentage of total donor Pmel-1 cells (**d**) and KLRG1$^{hi}$ Pmel-1 cells (**e**) in circulating PB from each group of mice receiving GFP-, Ezh2- and Ezh2$_{S21A}$-7 day-Pmel-1 cells over a period of 24 days after transfer. **f–h** In separate experiments as described in (**c–e**) above, donor T cells were recovered from the spleen and tumor at 21 days after transfer. **f** Plots show the percentage of transferred GFP+ cells in tumor and spleen by gating on donor Thy1.1+ Pmel-1 CD8+ cells, with Pmel-1 cells prior to transfer as controls. **g** Graphs show the number of total donor GFP+Thy1.1+ Pmel-cells and IFN-γ+ Pmel-1 cells in the spleen and tumor. **h** Plots show the expression of CD62L on the surface of donor GFP+Thy1.1+ T cells derived from the spleen and tumor. *$p < 0.05$, **$p < 0.01$, and ***$p < 0.001$ (two-tailed unpaired $t$ test). The data are pooled from two experiments (**a**; GFP, $n = 9$; Ezh2, $n = 9$; S21A, $n = 10$; mean ± SD), or representative of two independent experiments (**b**; $n = 3$-5 mice per group in each, mean ± SD; **c–h**; $n = 5$ mice per group in each, mean ± SD)

strategies to optimize the expansion and quality of therapeutic T cells for ACT.

Histone modifications occur in T cells early after antigen priming[17–19]. Recent studies suggested that CD8+ T-cells decreased the amount of H3K27me3 at the promoter regions of genes associated with cell proliferation and differentiation within 24 h[18]. We found that 3 days after TCR engagement Ezh2 was highly induced in CD8+ T cells, however, its function on activating *Id3* and repressing *Id2*, *Eomes* and *Prdm1* was decreased. It

has been shown that low expression of *Id3* and high levels of *Id2*, *Blimp-1* and *Eomes* are associated with enhanced effector differentiation but decreased memory potential[1,3,5,33,34,51,52]. In line with these observations, our data suggested that through coordinating the expression of these major TFs in activated CD8+ T cells, Ezh2 played essential roles in preserving T$_{CMP}$ and preventing precocious effector differentiation. We also observed that Ezh2 directly bound to the promoter region of *Id3* and activated *Id3* transcription in a dose-dependent manner. Introduction of

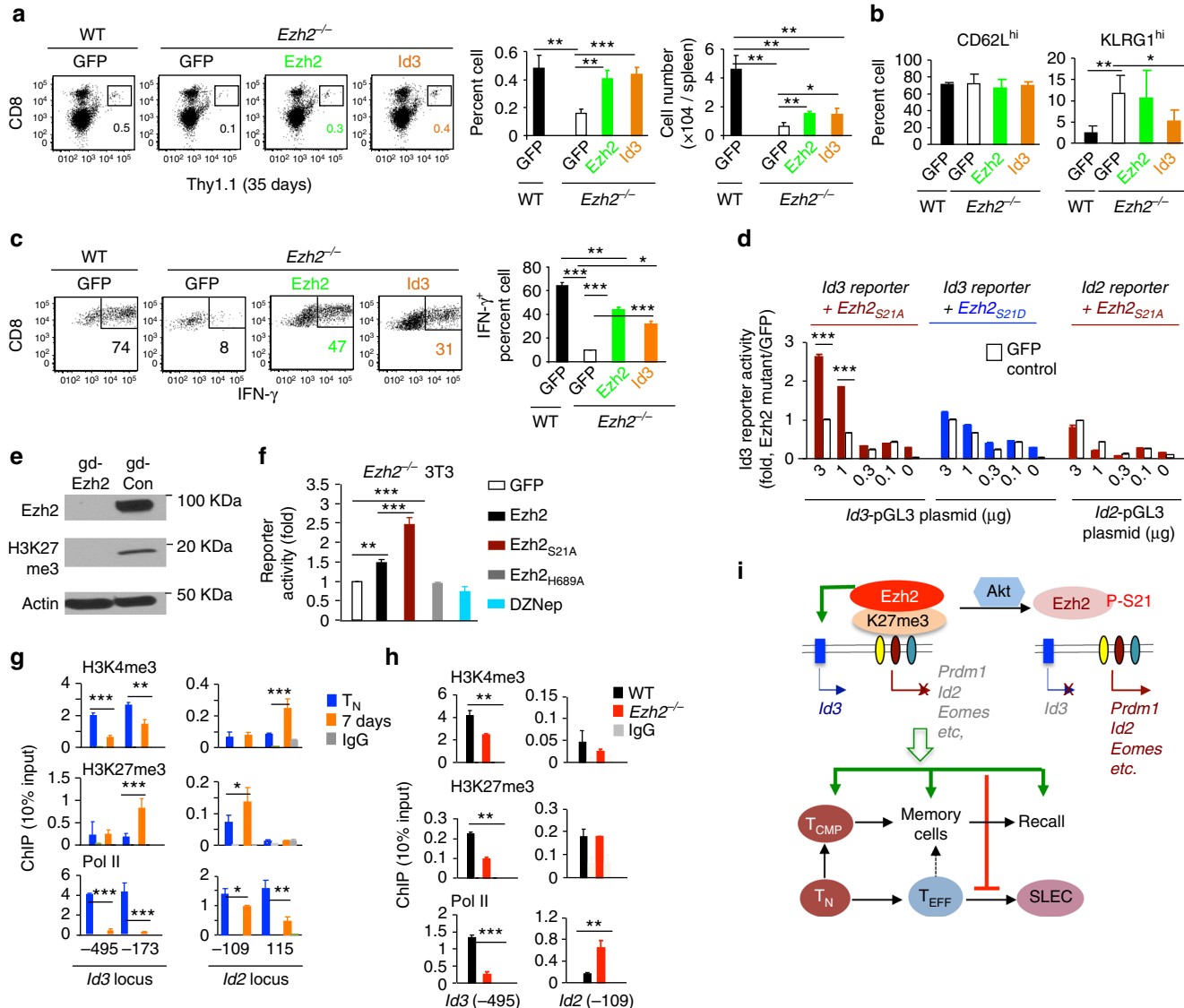

**Fig. 8** Id3 is a downstream effector of Akt-unphosphorylated Ezh2 in CD8[+] T cells. **a–c** WT and $Ezh2^{-/-}$ $T_N$ Pmel-1 cells (Thy1.1[+]) were activated and infected with MigR1 encoding GFP, Ezh2, and Id3, respectively. By 7 days, GFP[+] T cells were sorted and transferred into sublethally irradiated B6 mice (Thy1.2[+], $5 \times 10^5$ cells per mouse). IL-2 and gp100/DCs were administered to these mice for 3 days after the transfer. Donor T cells were collected from the spleen 35 days after transfer to measure their percentage and numbers (**a**), surface phenotype (**b**) and production of IFN-γ (**c**). Dot plots and graphs in **c** show the percent of IFN-γ[+] cells within donor Thy1.1[+] T cells. **d** 3T3 cells were transduced with an Id3-specific pGL3 luciferase reporter (Id3-pGL), together with MigR1-GFP, MigR1-Ezh2$_{S21A}$ or MigR1-Ezh2$_{S21D}$. Id2-specific pGL3 luciferase reporter (Id2-pGL) was used as control. Graph shows the fold change of luciferase reporter activity. The data (mean±SD) are representatives of three independent experiments. **e, f** 3T3 cells were transduced with pL-CRISPR.EFS.GFP plasmid or pL-CRISPR.EFS.GFP plasmid encoding guide RNA targeting Ezh2. 3 days later, GFP[+] cells were sorted into single cell using FACS sorter to select Ezh2-knockout cell clones. Immunoloblots show the loss of Ezh2 in 3T3 cells with Ezh2 knockout (**e**). These Ezh2-null 3T3 cells were reconstituted with MigR1-GFP vector and MigR1 encoding Ezh2, Ezh2$_{S21A}$ or Ezh2$_{H689A}$. Id3-pGL luciferase reporter was introduced to these cells for reporter assay. In some experiments, WT 3T3 cells were treated with DZNep to deplete Ezh2 protein. Graph shows the fold change of luciferase reporter activity (**f**). **g** WT Pmel-1 cells were stimulated with anti-CD3/CD28 Ab + IL-2 for 7 days for ChIP analysis. Graphs show the deposition of H3K4me3, H3K27me3 and Pol II at the promoter regions of Id3 and Id2. **h** WT and $Ezh2^{-/-}$ $T_N$ Pmel-1 cells (Thy1.1[+]) were stimulated with anti-CD3/CD28 Ab + IL-2 for 7 days to perform ChIP analysis. **i** Graphic summary of Ezh2 effects on memory formation and function. $*p < 0.05$, $**p < 0.01$, and $***p < 0.001$ (two-tailed unpaired $t$ test). The data are representatives of two experiments with $n = 3–4$ mice per group in each (**a–c**, mean ± SD), or three independent experiments (**d–f**, mean ± SD)

Id3 into activated CD8[+] T cells lacking Ezh2 rescued their recall response capacity for producing effector cells upon antigen reencounter. These data identify the importance of Ezh2 in regulating Id3-mediated memory T-cell development and function.

The results from our studies establish that Akt-mediated Ezh2 phosphorylation as a critical target for optimizing the expansion and quality of therapeutic T cells used for treating cancer and chronic infections. The capability of antigen-specific T cells to replicate and persist in vivo is crucial for controlling tumor growth and eliminating virus-infected cells[4,7,8,11,32,41–43]. In support of this, we have identified that Akt-unphosphorylated Ezh2-mediated epigenetic effects are involved in regulating the quantity and quality of memory precursors. These data support the observation that ex vivo treatment of expanding CD8[+] T cells

with certain Akt inhibitors led to increased frequency of memory precursors and improved antitumor immunity in vivo[38,39]. However, ex vivo Akt inhibitor treatment resulted in a transient effect and did not prevent reactivation of Akt in CD8[+] T cells in vivo upon antigen rechallenge. When signals activating Akt persist, the prolonged presence of Akt-insensitive Ezh2 in tumor-reactive CD8[+] T cells is required for them to preserve their memory properties in vivo. Continual treatment with Akt inhibitors in vivo might extend its effect on promoting memory T cells. However, effective cancer immunotherapy requires collective efforts of effector and memory T cells[4,6,11]. Systemic administration of Akt inhibitors may potentially inhibit effector responses and cause adverse effects. Future investigations will be identifying new pharmacological approaches that can selectively block the Ezh2 and Akt interaction in T cells for improving antitumor efficacy.

Our findings together with Kakaradov's studies[53] also indicate that the impact of Ezh2 on antigen-driven CD8[+] T cells may vary at different activation state and differentiation stage. Ezh2 promoted the survival and expansion of activated CD8[+] T cells later after antigen priming. By 5 days after activation, Ezh2 deficiency selectively increased apoptosis in $T_{EFF}$ rather than $T_{CM}$-like cells[53]. However, early after antigen activation (e.g., by 3 days) Ezh2 was dispensable for survival of antigen-primed CD8[+] T cells[53]. We also identified that loss of Ezh2 caused preferential decrease of $T_{CMP}$ pool independent of cell apoptosis early during expansion, while markedly increasing the fraction of terminal $T_{EFF}$. Since both the first division of activated CD8[+] T cells and their subsequent differentiation may influence the ratio of $T_{CMP}$ and $T_{EFF}$[46,47,53,54], we propose that Ezh2 is important for preserving the $T_{CMP}$ pool and restraining precocious terminal differentiation. Interestingly, Kakaradov's studies showed that despite the presence of Ezh2, about 75% of P14 CD8[+] T cells had undergone terminal differentiation (e.g., expression of KLRG1[hi]) at 4 days after LCMV infection and only 1% of activated P14 CD8[+] T cells were $T_{CMP}$-like phenotype 7 days after infection[53]. In contrast, we found that WT Pmel-1 cells consisted of <10% KLRG1[hi] cells and ~50% $T_{CMP}$ at the peak of effector response. Loss of Ezh2 led to production of 4-fold more terminal KLRG1[hi] cells, accompanied with increased expression of 370-fold more p19[Arf] transcript. These differences between our studies and Kakaradov's are likely explained by different experimental systems and the possibility that LCMV-induced excessive terminal differentiation could mask certain impact of Ezh2 in CD8[+] T cells. However, both studies support previous observations that $T_N$ are able to develop into memory cells without transitioning through an effector stage[54–63], in which Ezh2 plays a crucial role.

In conclusion, Ezh2 plays multiple roles in antigen-driven CD8[+] T-cell responses (Fig. 8i). Ezh2 promotes the survival and expansion of proliferating effector cells. Ezh2 also functions as a key molecular gatekeeper for generating CD8[+] T memory precursors, restraining terminal differentiation and maintaining antigen-specific recall response capability. Ezh2 activates *Id3* to regulate the persistence and function of effector and memory CD8[+] T cells. Ezh2 also silences *Id2*, *Prdm1* and *Eomes* to temper effector differentiation. Importantly, Ezh2 itself is functionally modified by Akt phosphorylation, which diminishes the capacity for Ezh2 to stimulate *Id3* and silence *Id2*, *Prdm1* and *Eomes*, thereby driving effector differentiation and reducing memory potential. Our findings also open new perspective to understand how extrinsic signals from environmental cues may influence T-cell immunity via modulating Ezh2 function.

## Methods

**Mice.** C57BL/6 (B6, Thy1.2[+]), and transgenic Pmel-1 (B6.Cg-*Thy1*[a]/Cy Tg (TcraTcrb)8Rest/J, Thy1.1[+]) mice were purchased from The Jackson Laboratories.

Ezh2 was deleted in gp100-specific CD8[+] T-cell receptor -transgenic Pmel-1 cells by backcrossing Ezh2[fl/fl] CD4-Cre B6 mice to Pmel-1 mice to produce T-cell-specific Ezh2-knockout Pmel-1 mice (Ezh2[−/−] Pmel-1). Both female and male mice (6–12 weeks of age) were used for experiment. Experimental protocols were approved by the Temple University's Committee on Use and Care of Animals, and the University of Michigan Committee on Use and Care of Animals.

**Cell preparation and culture.** CD44[lo]CD8[+] $T_N$ cells were isolated from spleen and LN (purity usually 90–95%). T cells were cultured in the presence of anti-CD3 (2 µg/ml; BD bioscience) and anti-CD28 (2 µg/ml; BD bioscience) Abs together with recombinant human IL-2 (10 ng/ml; R&D Systems). In vivo sorted T cells and cultured T cells were restimulated with anti-CD3 Ab (1 µg/ml) or human gp100 (10[−6] M) for 5 h before intracellular staining. BM-derived c-kit + cells were used to generate DCs. For TAT-Cre experiments, 10[6] purified CD8[+] T cells or splenocytes were incubated in 100 µl serum-free RPMI containing 90–100 µg/ml TAT-Cre fusion protein (a gift from Dr. Warren Pear, University of Pennsylvania) for 16 h at 37 °C. After extensive washing, treated cells were kept in vitro culture for another 5 days.

**Retroviral construction and T-cell infection.** MigR1 retroviral vector was provided by Dr. Warren Pear (University of Pennsylvania). Ezh2, Ezh2 mutants, and Id3 cDNA was cloned into MigR1 (GFP) vector. For retroviral infection, CD8[+] T cells were prestimulated with anti-CD3/CD28 Ab for 24 h, and then the retrovirus supernatant was added in the presence of 8 µg/ml polybrene (Sigma). Cells were spinoculated at 3000 r.p.m., 32 °C for 3 h. The same retroviral infection procedure was repeated 24 h later[23].

**Adoptive cell transfer, infection and tumor challenge.** Pmel-1 T cells (1 × 10[4] to 1 × 10[6]) were transferred into sublethally irradiated B6 mice (5 Gy) or non-irradiated B6 mice. IL-2 (100,000 IU) was administered intraperitoneally twice a day, and BM-derived DCs that had been pulsed with gp100 (10 µg) were transferred via tail vein, for constitutively three days after adoptive transfer. B6 mice were injected subcutaneously with 1 × 10[5] B16 melanoma cells. In some experiments, Ezh2[fl/fl] mice were intravenously received VVA-gp100 or VVA-OVA.

**Counting of adoptively transferred cells.** Mice were killed after infection. Samples were enriched for mononuclear cells or CD8[+] T cells (Miltenyi), and cells were counted by trypan blue exclusion. The frequency of transferred T cells was determined by measurement of the expression of CD8 and Thy1.1 by flow cytometry. The absolute number of Pmel-1 cells was calculated by multiplication of the total cell count with the percentage of CD8[+] Thy1.1.

**RNA-Seq analysis.** RNA sequencing was carried out at the sequencing core at University of Michigan Sequencing Core (Ann Arbor, Michigan). Transcriptome analysis was performed on RNA isolated from fresh naive and cultured CD8[+] T cells. Briefly, total RNA was isolated from T cells using an RNAeasy kit (QIAGEN) and RNA-seq libraries were prepared using SureSelect RNA Library Preparation kits (Agilent Technologies). Samples were run on a HiSeq 2000 sequencing system (Illumina), and at least 37.5 × 10[6] single-end reads were obtained per sample. Expression was evaluated by determining the fragment per kilobase per million reads values. Using one-way ANOVA analysis, we selected transcripts with $p < 0.01$ and $q < 0.01$ for comparing paired groups and at least a 1.5-fold difference from the means for the paired groups. RNAseq data were deposited in the NCBI's Gene Expression Omnibus database (accession no. GSE76755).

**Reverse transcription polymerase chain reaction (RT-PCR).** RNA was isolated with an RNeasy Mini kit (Qiagen) and cDNA was generated by reverse transcription (Applied Biosystems). Real-time RT-PCR was performed with a SYBR green PCR mix (ABI Biosystems) in the Realplex Eppendorf Real-time PCR instrument (Eppendorf AG). Gene expression levels were calculated relative to the *18s* gene. Data were collected and quantitatively analyzed on a Realplex sequence detection system (Eppendorf AG), and Applied Biosystems StepOne Plus Real-time PCR systems (Applied Biosystems). The primer sequences are listed in Supplementary Table 1.

**ChIP.** A Millipore ChIP kit and Diagenode ChIP kit was used for ChIP assay. DNA-protein complexes were crosslinked with formaldehyde at a final concentration of 1%. Sonicated extracts were precleared and incubated with Abs specific to Ezh2, H3K27me3, or nonspecific anti-IgG. The immunoprecipitated DNA was quantified by real-time quantitative PCR. The primer sequences are listed in Supplementary Table 2. The Ab information is listed in Supplementary Table 3.

**Western blot analysis.** Cell lysates were prepared in lysis buffer (150 mM NaCl, 50 mM Tris-Cl, 1% Triton X-100, 0.1% SDS, 0.5% sodium deoxycholate, 0.02% sodium azide, 1 mM sodium vanadate, protease inhibitors (10 µg/ml leupeptin, 10

μg/ml aprotinin, and 1 mM phenylmethylsulfonyl fluoride)). Protein concentrations were determined using a Bio-Rad protein assay. Ten to 50 μg protein was loaded in 4–12% Mini-PrROTEAN TGX Precast protein Gels (Bio-rad), and transferred to PVDF membrane. Membranes were blocked in 3% BSA for 1 h, incubated with the primary antibody overnight at 4 °C, washed, and incubated with the appropriate horseradish-conjugated secondary antibody for 1 h at room temperature. An enhanced chemiluminescent (ECL) kit was used to visualize the signal (Thermo Fisher). The Abs used for western blot analysis are listed in Supplementary Table 3. Uncropped blots for western blot analysis are shown in Supplementary Fig. 11.

**Flow cytometric analysis and cell lines**. The Abs used for flow cytometric analyses were purchased from eBioscience, BioLegend, or BD Biosciences. The Ab information is listed in Supplementary Table 3. Flow cytometric analyses were performed with FACS LSRII (BD Biosciences) as described[64,65]. B16 mouse melanoma cell line was purchased from American Type Culture Collection, and were cultured in RPMI-1640 medium (Gibco) supplemented with 10% FBS. No mycoplasma contamination was detected in any of the cultures using a mycoplasma detection kit.

**Statistical analyses**. Unless otherwise specified, statistical tests were performed using unpaired two-tailed Student's $t$ test. Where necessary, the Shapiro−Wilk test was used to test for normality of the underlying sample distribution. No blinding was done, as objective quantitative assays such as flow cytometry, were used. Experimental sample sizes were chosen using power calculations with preliminary experiments and/or were based on previously described variability in similar experiments[30,45,50,65–67]. Samples that had undergone technical failure during processing were excluded from analyses. Where relevant, recipient mice were randomized before adoptive transfer. $p$ values of 0.05 or less were considered significant.

**Data availability**. The RNA-seq data has been deposited in National Center for Biotechnology Information Gene Expression Omnibus (GEO)database (https://www.ncbi.nlm.nih.gov/geo) under the accession code GSE76755. All other materials are available from the corresponding authors upon reasonable request.

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

## Acknowledgements

We thank Drs Stephen Emerson, Elizabeth Hexner and Ananda Goldrath for helpful comments. We thank Dr. Daohai Yu (Temple University) for statistical analyses and evaluation. This study is supported by grants of ACS (Y.Z.), DOD (Y.Z.), NCI (CA172106-01, Y.Z.), NHLBI (HL127351-01A1, Y.Z.) and Fels Pilot Grant (S.H.).

## Author Contributions

S.H. and Y.Z. conceived and designed the project; S.H., Y.L., LM, H.S., J.P., P.C., C.H., and Y.L performed experiments and analyzed the data; B.M., S.H., L.G., and Y.Z, designed experiments and analyzed data; J.M. analyzed RNA-seq data; L.G., J.S., and R.R. edited the manuscript; and S.H. and Y.Z. wrote and edited the manuscript.

## Additional information

**Competing interests:** The authors declare no competing financial interests.

