## [Peer Review File · Nature Communications]

Reviewers' comments:

Reviewer #1 (Remarks to the Author):

In this manuscript, entitled 'The Phosphorylation State of Ezh2 Determines its Capacity to Maintain CD8+ Memory T Cells for Antitumor Immunity', He et al. provide extensive evidence for a novel role of Ezh2 in regulation of CD8+ T cell differentiation. The authors have gone to substantial lengths to respond to my original comments and have now provided satisfactory experimental evidence for most of the issues raised. Evidence of biased/directional involvement of histone-modifying enzymes in specific T cell differentiation programmes and signalling processes is both timely and important to the field. In my opinion, the manuscript is acceptable for publication, once two remaining points have been addressed:

1. To fully evidence the role of Ezh2 phosphorylation in regulation of T cell differentiation, I feel an adequate response to my previous comment #6 should be provided. In revised Fig 6e, the correct control for overexpression of either S21A or S21D Ezh2 is overexpression of WT Ezh2. Currently, phospho-mutant Ezh2 overexpressing cells are compared with GFP-transduced cells, but not WT Ezh2 overexpressing cells. Thus, it is not clear whether S21 phosphorylation is causing the effects seen in the figures or mere overexpression of Ezh2. It would be helpful to directly compare the effect of WT and S21 mutant Ezh2 overexpression in the same experiment, as has now been done in Fig 5i. It's fine if the overexpression system is reducing the opportunity to see a 'gain of function' caused by the alanine mutation compared with WT - since this can be explained as a result of overexpression and could be discussed in the Discussion. However, the data should be included.

2. Could the authors compare their findings and conclusions with the recent paper describing Ezh2 function in CD8+ T cells by Kakaradov et al in Nature Immunology (2017)? Is the proposed function of Ezh2 similar or different, and why? A short discussion of this point would be sufficient.

Reviewer#3 (Remarks to the Author):

I believe that the authors have mostly addressed the questions of the referee#2 with additional experiments or explanations. This is an extensive amount of work (also in response to reviewer #1).

However, I still do not believe that any of the new or old experiments directly address the main premise of the paper that Ezh2 controls memory CD8 T cell formation. This is because as mentioned by the reviewer there appears to be such a strong effect of Ezh2 deficiency on the acute response it makes it very difficult to interpret the effects on memory generation. The use of transgenic T cells I believe can lead to misinterpretation of the role of Ezh2 in CD8 T cells for a number of reasons. I therefore am asking for one simple experiment that should address this question.

I would like the authors to challenge lymphoreplete WT and Ezh2^{fl/fl} CD4^{cre} (Ezh2^{-/-}) mice and to track the endogenous antigen-specific CD8 T cells during the acute (at the peak of the response; usually around day 7) and memory CD8 T cell response. This would preferably be done using the VVA-gp100 for consistency but I would be happy with other means (such as DCs loaded with peptide antigen)

If the expansion of the Ezh2 deficient CD8 T cells in the primary response is indeed relatively normal (as claimed by the authors using the transgenic T cells) or there is a selective loss of memory precursors then I am happy to accept that Ezh2 is involved in memory CD8 T cell formation.

Reviewer #1 (Remarks to the Author): In this manuscript, entitled 'The Phosphorylation State of Ezh2 Determines its Capacity to Maintain CD8+ Memory T Cells for Antitumor Immunity', He et al. provide extensive evidence for a novel role of Ezh2 in regulation of CD8+ T cell differentiation. The authors have gone to substantial lengths to respond to my original comments and have now provided satisfactory experimental evidence for most of the issues raised. Evidence of biased/directional involvement of histone-modifying enzymes in specific T cell differentiation programmes and signalling processes is both timely and important to the field. In my opinion, the manuscript is acceptable for publication, once two remaining points have been addressed:

1. To fully evidence the role of Ezh2 phosphorylation in regulation of T cell differentiation, I feel an adequate response to my previous comment #6 should be provided. In revised Fig 6e, the correct control for overexpression of either S21A or S21D Ezh2 is overexpression of WT Ezh2. Currently, phospho-mutant Ezh2 overexpressing cells are compared with GFP-transduced cells, but not WT Ezh2 overexpressing cells. Thus, it is not clear whether S21 phosphorylation is causing the effects seen in the figures or mere overexpression of Ezh2. It would be helpful to directly compare the effect of WT and S21 mutant Ezh2 overexpression in the same experiment, as has now been done in Fig 5i. It's fine if the overexpression system is reducing the opportunity to see a 'gain of function' caused by the alanine mutation compared with WT - since this can be explained as a result of overexpression and could be discussed in the Discussion. However, the data should be included.

Our response and revision: We agree with the reviewer that it is important to differentiate the effects of Ezh2 overexpression from the effects of Ezh2 phosphorylation. Interestingly, overexpressing WT Ezh2 in *Ezh2*^{-/-} Pmel-1 cells rescued survival *in vivo* similar to Ezh2_{S21A} (Fig.6d), but, was significantly less effective at maintaining Id3, Prdm1 and Eomes expression (Fig.6c) and failed to sustain Id3^{hi} T cells (Fig.6e). Therefore, these investigations reveal critical contributions of Ezh2 phosphorylation for control of T memory cell differentiation, beyond T cell survival. This is added in our revised manuscript page 14, paragraph 1.

2. Could the authors compare their findings and conclusions with the recent paper describing Ezh2 function in CD8+ T cells by Kakaradov et al in Nature Immunology (2017)? Is the proposed function of Ezh2 similar or different, and why? A short discussion of this point would be sufficient.

Our response and revision:

As reviewer suggested, we carefully examined the similarity and difference in Ezh2 function between our studies and Kakaradov's. While their studies showed that Ezh2 bound genes identified by scRNA-seq in proliferating cells, they did not show the functional relevance of Ezh2 and Ezh2 deficiency on the expression of these genes in T cells. Most importantly, our findings indicate, as revealed by our RNA-seq analysis and extensive functional analysis, that Ezh2 can not only repress gene expression, but also activates gene transcription in antigen-driven T cells. For these reasons, we could not discuss the similarity and difference in function of Ezh2 deficiency on gene expression between our studies and theirs.

In stead, we focus on discussing the effect of Ezh2 on T cell responses at the cellular level, and add the following paragraph to the discussion section, page 22: Our findings together with Kakaradov's studies¹ indicate that the impact of Ezh2 on antigen-driven CD8⁺ T cells may vary at different activation states and stages of differentiation. Ezh2 was crucial for the survival and expansion of activated CD8⁺ T cells later after antigen priming. By day 5 after activation, Ezh2 deficiency selectively increased apoptosis in effector T cells rather than T_{CM}-like cells¹. However, early after antigen priming (e.g., by day 3) Ezh2 was dispensable for survival of antigen-primed CD8⁺ T cells.¹ Importantly, we

also identified that loss of Ezh2 caused a preferential decrease of the T_{CMP} pool independent of cell apoptosis early during expansion phase, while markedly increasing the fraction of terminal T_{EFF} differentiation. Since both the first division of activated $CD8^+$ T cells and their subsequent differentiation early after antigen priming may influence the ratio of T_{CMP} and T_{EFF} ^{1, 2, 3, 4}, we propose that Ezh2 is important for preserving the T_{CMP} pool and restraining precocious terminal differentiation. Interestingly, Kakaradov's studies showed that Ezh2 deficiency caused no difference in the production of terminal KLRG1^{hi} cells. They observed that despite the presence of Ezh2, as many as 75% of P14 $CD8^+$ T cells had undergone terminal differentiation (e.g. expression of KLRG1^{hi}) at day 7 after LCMV infection and only 1% of activated P14 $CD8^+$ T cells¹. In contrast, we found that WT Pmel-1 cells consisted of less than 10% KLRG1^{hi} cells and approximately 50% T_{CMP} at the peak of effector response. Loss of Ezh2 led to production of 4-fold more terminal KLRG1^{hi} cells, accompanied with increased expression of 370-fold more p19^{Arf} transcript. These differences between our studies and Kakaradov's could be explained by different experimental systems. In addition, in their studies the excessive terminal differentiation in response to LCMV infection could mask certain impact of Ezh2 in T cells¹.

Reviewer#3 (Remarks to the Author):

I believe that the authors have mostly addressed the questions of the referee#2 with additional experiments or explanations. This is an extensive amount of work (also in response to reviewer #1).

However, I still do not believe that any of the new or old experiments directly address the main premise of the paper that Ezh2 controls memory CD8 T cell formation. This is because as mentioned by the reviewer there appears to be such a strong effect of Ezh2 deficiency on the acute response it makes it very difficult to interpret the effects on memory generation. The use of transgenic T cells I believe can lead to misinterpretation of the role of Ezh2 in CD8 T cells for a number of reasons. I therefore am asking for one simple experiment that should address this question.

I would like the authors to challenge lymphoreplete WT and Ezh2^{fl/fl} CD4^{cre} (Ezh2^{-/-}) mice and to track the endogenous antigen-specific CD8 T cells during the acute (at the peak of the response; usually around day 7) and memory CD8 T cell response. This would preferably be done using the VVA-gp100 for consistency but I would be happy with other means (such as DCs loaded with peptide antigen). If the expansion of the Ezh2 deficient CD8 T cells in the primary response is indeed relatively normal (as claimed by the authors using the transgenic T cells) or there is a selective loss of memory precursors then I am happy to accept that Ezh2 is involved in memory CD8 T cell formation.

Our response and revision: We agree with the reviewer that a strong effect of Ezh2 deficiency on the acute response makes it very difficult to interpret the effects memory generation. In particular, we recognize that increased apoptosis in antigen-driven $CD8^+$ T cells in Ezh2-null cells later during expansion phase could be misinterpreted. Therefore, as the reviewer suggested, we performed new experiments to track the endogenous antigen-specific $CD8^+$ T cells during acute and memory $CD8^+$ T cell responses.

Studying endogenous antigen-specific $CD8^+$ T cell responses using Ezh2^{fl/fl} CD4^{cre} (Ezh2^{-/-}) mice could be potentially problematic because in these mice Ezh2 is also deleted in $CD4^+$ T cells. Since $CD4^+$ T cells are crucial for regulating the development and maintenance of $CD8^+$ memory T cells, it would be impossible to dissect the $CD8^+$ intrinsic and $CD4$ -dependent contribution of Ezh2 deletion in $CD8^+$ T cell memory. To avoid these caveats and more stringently examine the cell autonomous effect

of Ezh2 on endogenous CD8⁺ T cells with the presence of WT CD4⁺ T cells, we co-transferred equal amount of WT (Thy1.1⁺CD45.2⁺) and Ezh2^{-/-} (Thy1.1⁺CD45.2⁺) splenocytes into lymphoreplete B6/SJL mice (Thy1.1⁻CD45.1⁺) and subsequently challenged the recipient mice with a vaccinia virus encoding ovalbumin (VVA-OVA).

Our new experiments confirm a specific role for Ezh2 in T cell memory formation and differentiate this function from cell survival, since loss of Ezh2 leads to earlier occurrence of the peak of T cell response and preferential loss of memory precursors. As compared to WT CD8⁺ T cells, Ezh2^{-/-} CD8⁺ T cells produced 1.5- to 2-fold more OVA₂₅₇₋₂₆₄-specific T cells 3 days after infection, maintained at day 4, and dramatically declined by day 5 (**Supplementary Fig. 4a,b**). Notably, the increase of OVA₂₅₇₋₂₆₄-specific Ezh2^{-/-} CD8⁺ T cells 3 days after infection was associated significantly enhanced proliferation rates (**Supplementary Fig. 4c**), increases of KLRG1^{hi} cells (**Supplementary Fig. 4c,d**) and selectively decreased ratio of T_{CMP} versus T_{EFF} (**Supplementary Fig. 4f**). These new results are discussed in paragraph 1, page 8.

To explain the important effect of antigen priming on the generation of memory precursors early during primary response, we add the following discussion in Page 22: “.....Since both the first division of activated CD8 T cells and their subsequent differentiation early after antigen priming may influence the ratio of T_{CMP} and T_{EFF}^{1, 2, 3, 4}, we propose that Ezh2 is important for preserving the T_{CMP} pool and restraining precocious terminal differentiation.....”

Again, we observed that 30 after infection, loss of Ezh2 induced 5-folds less memory T cells (5.7% ± 0.6 versus 1.1 % ± 0.6, P<0.0001). These memory T cells lacking Ezh2 showed markedly inferior capacity to produce high levels of IFN-γ compared to their WT counterparts (22.3% ± 9.0 versus 53.7% ± 9.6, P<0.05). This result is not included in the revision to reduce the lengthy of this manuscript.

Reference:

1. Kakaradov, B. *et al.* Early transcriptional and epigenetic regulation of CD8⁺ T cell differentiation revealed by single-cell RNA sequencing. *Nature immunology* **18**, 422-432 (2017).
2. Chang, J.T. *et al.* Asymmetric T lymphocyte division in the initiation of adaptive immune responses. *Science* **315**, 1687-1691 (2007).
3. Gerlach, C. *et al.* Heterogeneous differentiation patterns of individual CD8⁺ T cells. *Science* **340**, 635-639 (2013).
4. Buchholz, V.R. *et al.* Disparate individual fates compose robust CD8⁺ T cell immunity. *Science* **340**, 630-635 (2013).

Reviewer #1 (Remarks to the Author):

In their article, 'The Phosphorylation State of Ezh2 Determines its Capacity to Maintain CD8+ Memory T Cells for Antitumor Immunity,' He et al have provided evidence for the contextual function of Ezh2 in CD8+ T cell differentiation. Evidence of biased/directional involvement of histone-modifying enzymes in T cell differentiation and their relationship with signalling processes is both timely and important to the field, especially given the amenability of such mechanisms to pharmacological manipulation. In my opinion, the authors have gone to extensive lengths to address my previous comments and the manuscript is now acceptable for publication.

Reviewer #3 (Remarks to the Author):

The authors performed a well planned experiment to address my concern that loss of ezh2 in T cells simply leads to a massively reduced primary response and not a specific memory T cell development defect.

However, the experiment the authors set up to answer my question appears to go against the overall premise of the paper that Ezh2 controls memory T cell development. It appears to support the idea that the majority of the Ezh2 dependent effects on the CD8 T cell response are in the initial generation of the response leading to massive loss of cells between days 5 and 7.

I am not happy with this because it says that it is most likely not a Ezh2-dependent "programming" of memory T cells but simply a dramatic loss of the effector pool (and not a selective loss of memory precursors-see day 5 of Figure S4f).

Reviewer-3

The authors performed a well planned experiment to address my concern that loss of *ezh2* in T cells simply leads to a massively reduced primary response and not a specific memory T cell development defect.

However, the experiment the authors set up to answer my question appears to go against the overall premise of the paper that Ezh2 controls memory T cell development. It appears to support the idea that the majority of the Ezh2 dependent effects on the CD8 T cell response are in the initial generation of the response leading to massive loss of cells between days 5 and 7.

I am not happy with this because it says that it is most likely not a Ezh2-dependent "programming" of memory T cells but simply a dramatic loss of the effector pool (and not a selective loss of memory precursors-see day 5 of Figure S4f).

Our response and revision: Based on our data showing no significant difference in the frequency of T_{CMP} at day 5 after infection between WT and *Ezh2*^{-/-} T cells (**Fig.S4f**), the reviewer concluded "it is most likely not a Ezh2-dependent "programming" of memory T cells but simply a dramatic loss of the effector pool".

We appreciate the reviewer's critique, which stimulates us to further clarify this point in our revision as described below.

A clear understanding of Ezh2 effects on memory precursors has profound implications for the development of effective ACT for cancer and chronic infections. Although some reviews have reinforced the textbook view that memory T cells arise from effector cells,¹ emerging evidence suggest that T_N may directly develop into memory cells without transitioning through an effector stage. Within the first 1-2d of acute infection, the expansion, acquisition of effector function, contraction and the development of long-lived memory CD8⁺ T cells are largely programmed^{2, 3, 4, 5, 6, 7, 8, 9, 10, 11}. Using various models of immune responses, we consistently showed that Ezh2 deficiency led to selective loss of memory precursors within 4d of activation. During this period, Ezh2-deficiency caused significant and selective reduction of T_{CMP} , increase of terminally differentiated KLRG-1^{hi} T cells, and enhanced proliferation rates of endogenous antigen-specific T cells (Fig.S4c). Most importantly, it is likely that modification of Ezh2 function in antigen-driven CD8⁺ T cells might explain why their fate of becoming memory cells is largely programmed early after antigen activation. We observed that proliferating T cells actively modified Ezh2 function via Akt-mediated phosphorylation of Ezh2. Following antigen activation, WT T cells showed significant increases in pEzh2_{S21} at d3 and further increased by d5. Phosphorylation of Ezh2 also altered Ezh2 function in both T_{CMP} and T_{EFF} , with more dramatic changes in T_{EFF} . Overexpression of mutant Ezh2_{S21A} significantly increased the fraction of T_{CMP} while decreasing terminally differentiated KLRG-1^{hi} cells compared to WT Ezh2. These data substantially support the notion that Akt-unphosphorylated Ezh2 is critical for preserving the T_{CMP} pool. This may explain why there was a significant reduction of WT T_{CMP} at d5 compared to d3, as well as diminished difference in frequency of T_{CMP} between WT and *Ezh2*^{-/-} cells at d5 (Fig.S4f). Besides its quantitative effect on T_{CMP} , Ezh2 influences the quality of T_{CMP} developed during primary immune response. T_{CMP} derived from activated *Ezh2*^{-/-} CD8⁺ T cells failed to produce mature memory T cells, despite their capability of surviving during memory differentiation (Fig.2h,i). Combined with all these observations, we suggest that our results unambiguously establish the role of Ezh2 in programming memory precursors during the first 3 days of activation.

References:

1. Antia, R., Ganusov, V.V. & Ahmed, R. The role of models in understanding CD8+ T-cell memory. *Nat Rev Immunol* **5**, 101-111 (2005).
2. Williams, M.A. & Bevan, M.J. Effector and memory CTL differentiation. *Annu Rev Immunol* **25**, 171-192 (2007).
3. Mercado, R. *et al.* Early programming of T cell populations responding to bacterial infection. *J Immunol* **165**, 6833-6839 (2000).
4. Prlic, M., Hernandez-Hoyos, G. & Bevan, M.J. Duration of the initial TCR stimulus controls the magnitude but not functionality of the CD8+ T cell response. *J Exp Med* **203**, 2135-2143 (2006).
5. Badovinac, V.P., Porter, B.B. & Harty, J.T. Programmed contraction of CD8(+) T cells after infection. *Nature immunology* **3**, 619-626 (2002).
6. Williams, M.A. & Bevan, M.J. Shortening the infectious period does not alter expansion of CD8 T cells but diminishes their capacity to differentiate into memory cells. *J Immunol* **173**, 6694-6702 (2004).
7. Henrickson, S.E. *et al.* Antigen availability determines CD8(+) T cell-dendritic cell interaction kinetics and memory fate decisions. *Immunity* **39**, 496-507 (2013).
8. Mempel, T.R., Henrickson, S.E. & Von Andrian, U.H. T-cell priming by dendritic cells in lymph nodes occurs in three distinct phases. *Nature* **427**, 154-159 (2004).
9. Chang, J.T. *et al.* Asymmetric T lymphocyte division in the initiation of adaptive immune responses. *Science* **315**, 1687-1691 (2007).
10. Restifo, N.P. & Gattinoni, L. Lineage relationship of effector and memory T cells. *Current opinion in immunology* **25**, 556-563 (2013).
11. Jung, S. *et al.* In vivo depletion of CD11c+ dendritic cells abrogates priming of CD8+ T cells by exogenous cell-associated antigens. *Immunity* **17**, 211-220 (2002).

Reviewer #3 (Remarks to the Author):

The authors have attempted to reconcile their new experiment (fig.s4) with an argument about early programming of the memory response. I am not convinced by this and I believe the authors would have to undertake a new series of experiments to support this speculation which may again not support their hypothesis.

I have no issues with the data and experiments in this paper but I have serious concerns about the interpretation that the effect is specifically on the generation of memory T cells. However the data clearly shows that Ezh2 is critical for the generation of CD8 T cell responses (both effector and memory).

I think the authors should consider reinterpreting their data towards a more general role for Ezh2 and Ezh2 phosphorylation on the CD8+ T cell response.

Reviewer #3 (Remarks to the Author): The authors have attempted to reconcile their new experiment (fig.s4) with an argument about early programming of the memory response. I am not convinced by this and I believe the authors would have to undertake a new series of experiments to support this speculation which may again not support their hypothesis.

I have no issues with the data and experiments in this paper but I have serious concerns about the interpretation that the effect is specifically on the generation of memory T cells. However the data clearly shows that Ezh2 is critical for the generation of CD8 T cell responses (both effector and memory).

I think the authors should consider reinterpreting their data towards a more general role for Ezh2 and Ezh2 phosphorylation on the CD8+ T cell response.

Our response and revision: We agree with the reviewer that Ezh2 has general roles in the regulation of antigen-driven CD8 T cell responses, including both effector and memory responses. We acknowledge this point in our revised manuscript as the following:

... “Ezh2 is crucial for persistence of effector and memory T cells under either lymphopenic or lymphoreplete conditions.” This is shown in page 6, the second paragraph.

... “Ezh2 is dispensable for the homeostatic survival of T_{CMP} during contraction phase, but is important for the transition of both T_{CMP} and T_{EFF} into mature memory T cells.” This is shown in page 8, the last paragraph.

... “we propose that Ezh2 orchestrates the expression of these TFs for controlling stepwise effector differentiation and memory formation of CD8⁺ T cells.” This is shown in Page 10, the first paragraph.

... “our observations that Ezh2 regulates effector, memory precursor and antigen-specific recall response have profound implications towards the development of new strategies to optimize the expansion and quality of therapeutic T cells for ACT.” This is shown in page 17, the first paragraph.

... “Ezh2 plays multiple roles in antigen-driven CD8⁺ T cell responses. Ezh2 promotes the survival and expansion of proliferating effector cells. Ezh2 also functions as a key molecular gatekeeper for generating CD8⁺ T memory precursors, restraining terminal differentiation and maintaining antigen-specific recall response capability (**Fig.8i**).” This is shown in page 19 (the last paragraph) and page 20 (the first paragraph).

Finally, to tone down our argument about the role of Ezh2 in memory T cells, we remove the paragraph stressing the role of Ezh2 in promoting memory precursor cell development. In stead, we add the following sentence at end of the first paragraph, page 19:“However, both studies support previous observations that T_N are able to develop into memory cells without transitioning through an effector stage,^{52, 53, 54, 55, 56, 57, 58, 59, 60, 61} in which Ezh2 plays a crucial role.”